# Integrative analysis of genomic and epigenomic regulation reveals miRNA mediated tumor heterogeneity and immune evasion in lower grade glioma
Zhen Yang [1,13] ✉, Xiaocen Liu[2,3,4,13], Hao Xu[5,6,13], Andrew E. Teschendorff [7], Lingjie Xu[8], Jingyi Li[9], Minjie Fu[5,6], Jun Liu[10], Hanyu Zhou[3,4,11], Yingying Wang[2], Licheng Zhang[5,6], Yungang He[12], Kun Lv [3,4,11] ✉ & Hui Yang [3,4,11] ✉

The expression dysregulation of microRNAs (miRNA) has been widely reported during cancer development, however, the underling mechanism remains largely unanswered. In the present work, we performed a systematic integrative study for genome-wide DNA methylation, copy number variation and miRNA expression data to identify mechanisms underlying miRNA dysregulation in lower grade glioma. We identify 719 miRNAs whose expression was associated with alterations of copy number variation or promoter methylation. Integrative multi-omics analysis revealed four subtypes with differing prognoses. These glioma subtypes exhibited distinct immune-related characteristics as well as clinical and genetic features. By construction of a miRNA regulatory network, we identified candidate miRNAs associated with immune evasion and response to immunotherapy. Finally, eight prognosis related miRNAs were validated to promote cell migration, invasion and proliferation through in vitro experiments. Our study reveals the crosstalk among DNA methylation, copy number variation and miRNA expression for immune regulation in glioma, and could have important implications for patient stratification and development of biomarkers for immunotherapy approaches.

Lower grade glioma (LGG) is the most common and aggressive central nervous system (CNS) tumor, which accounting for about 40% of all brain malignancies[1]. Most LGGs can be further classified according to specific type of cell with which they share histological features or molecular biomarkers, such as isocitrate dehydrogenase (*IDH*) mutation or $O^6$-methyl-guanine-DNA methyltransferase (*MGMT*) promoter methylation[2,3]. Despite tremendous progress has been made over the past decades for its early detection, surgical paradigm and multidisciplinary treatment, such as neoadjuvant chemotherapy and radiotherapy, the postoperative overall survival of LGG patients remains extremely low, particularly for high-grade glioma with worse prognoses for their malignant aggressivity[4]. In general, the prognosis of patients with glioma varies dramatically, which is dependent on different tumor grade, key gene alternation status, tumor micro-environment, and combination of different efficacious methods[5]. Recent genomic studies have brought a comprehensive view of glioma based on molecular profiling and identified different subtypes within the glioma that appear to correlate with disease etiology and therapy response[6]. Hence, there

is an urgent need to characterize more specific and precise molecular signatures of LGG for its accurate diagnosis, individualized treatment and the prognosis assessment.

MicroRNAs (miRNAs) are ~22 nt small non-coding RNAs that play an important role in post-transcriptional regulation, which precisely adjust the gene expression level by targeting mRNAs for its degradation or translational repression[7,8]. miRNAs are critical for normal development of organisms and are involved in a variety of biological processes including cell cycle, cell proliferation, differentiation, apoptosis and cellular signaling[9,10]. Aberrant expression of miRNAs is associated with many human diseases[11,12]. For glioma, miRNA expression has been associated with the clinical and molecular classification[13], progression[14], prognosis[15] and response to targeted therapies[16–18]. Altered miRNA regulation is widely involved in glioma pathogenesis via the modulation of oncogenes and the associated downstream signaling pathways. Till now, most studies were proposed concerning aberrant expression and functional curation of miRNAs in cancer and disease, but there still lacks systematic investigation of the

mechanisms underlying miRNA regulatory activities, especially the genome-wide studies for genetic and epigenetic regulation of miRNAs in development and progression of glioma are still scarce.

In recent years, large multi-omics studies greatly enhanced our understanding of disease dysregulation at genomic and epigenetic levels. Genome wide alternations such as copy number variations (CNVs) and DNA methylations (DNAm) are widely observed during tumorigenesis, which promote cancer progression[19,20]. As a key genomic regulator that contribute to gene expression dysregulation via modulating mRNA levels and by influencing transcriptional regulation, CNV is significantly correlated with certain pathological processes such as cancer[21]. Transcriptional disorders caused by CNV changes have been identified as driving events in LGG progression[22]. In addition, gene expression regulation via DNA methylation at the epigenetic level also exerts a crucial part in the behaviors of LGG. DNAm imbalances is able to affect the occurrence and development of LGG by defining different types of driver events, such as cell growth, proliferation, differentiation and tumor-immune symbiosis[23,24]. Genomic profiling studies indicated that these genomic and epigenomic dysregulations are highly heterogeneous and jointly regulate the occurrence and evolution of tumor[25]. Till now, the influence of CNV and DNAm on noncoding RNA, especially miRNA expression remains largely unexplored. Studies indicated various regulators are required for precise control the expressions of miRNAs[26]. Therefore, miRNAs with aberrant expression under genetic and epigenetic contexts may serve as valuable source for investigation of the complex regulatory mechanisms in tumor progression, and a systematic analysis of CNV and DNAm patterns of miRNAs in large scale cancer samples is critically needed.

In this study, we collected copy number variation, DNA methylation and miRNA expression profiles from the cohort of The Cancer Genome Atlas (TCGA) LGG patients, and identified miRNAs whose expression levels are regulated by genomic and/or epigenomic deregulation. In addition, by using the multi-omics integration study, we identified different molecular subtypes of LGG that are significantly associated with prognosis. Furthermore, specific miRNA biomarkers were proposed to drive the classification of these subtypes. By assessing the associations between immune cell infiltrations and immune pathways, we identified a subset of immune-related miRNAs, and several miRNAs are identified as immune evasion markers with high confidence. Our study developed strategies of multi-omics integration and characterized the genetic and epigenetic landscape of genes encoding miRNAs, and also provides a framework for identifying molecular biomarkers to guide the clinical treatment for glioma patients.

## Results

### miRNA deregulation mediated by copy number variation and DNA methylation

We obtained the genome-wide profiling data of copy number variation, DNA methylation, and miRNA expression for 500 LGG patients from TCGA. We first explored the global CNV characteristics of LGG on the whole genome and found that there is a concentrated copy number amplification on q32.1 of chr1, p11.2 of chr7 and 12q.14 of chr11, as well as copy number deletion on q37.3 of chr2, p21.3 of chr3, q23.31 of chr10, and q13.2 of chr9 (Supplementary Fig. 1). This copy number amplitude data was then mapped to miRNA loci to build copy number profiles for 2265 miRNAs. Next, we obtained DNA methylation profiling data for 1977 miRNAs by mapping 8825 450K array probes to miRNA promoter regions (Supplementary Data 1 and 2). The metaplot depicts that the methylation level around transcription start sites was significantly lower than that from gene body and intergenic regions for miRNAs (Supplementary Fig. 2).

To assess the global effects of genomic and/or epigenomic aberrations on miRNA expression, the correlation coefficients between CNV or DNAm with corresponding miRNA expression were calculated and then normalized by using Fisher's Z-transformation. It was found that the overall correlation coefficients between miRNA expression and CNV shifted to the right (skewness = 2.43, $p < 2.2e{-}16$, D'Agostino test), whereas that between miRNA expression and DNAm significantly shifted to the left (skewness =

$-1.11$, $p < 2.2e{-}16$, D'Agostino test) (Fig. 1a). This indicated the overall positive and negative regulatory effect on miRNA expression by CNV and DNAm, respectively, which is similar to that of protein coding genes. We next focus on the positively correlated miRNAs for CNV (CNV-miR) and the negatively correlated miRNAs for DNAm (DNAm-miR). A total of 351 CNV-miRs (Supplementary Data 3) and 541 DNAm-miRs (Supplementary Data 4) were identified at the thresholds of FDR < 0.05. Genome-wide landscape for the CNV-exp associations and DNAm-exp associations are depicted as Supplementary Fig. 3. Some top ranked CNV-miRs and DNAm-miRs with significant associations are presented as Supplementary Figs. 4 and 5, respectively. Many of these miRNAs have been identified that play a key role in different cancers. One of the examples is the hsa-miR-26a, whose expression is highly correlated with CNV value ($r = 0.62$, $p < 2.2e{-}16$). This miRNA has been confirmed as a key driver gene of glioblastoma for the copy number amplification associated overexpression, which lead to the malignant transformation of cell and tumor growth promotion[27,28]. Another example is miR-155-5p, a strong negative correlation pattern was observed for its expression and promoter methylation ($r = -0.74$, $p < 2.2e{-}16$). This pattern has been confirmed by several other glioma related studies[29,30]. Similar phenomenon was also observed in many other cancers including breast cancer[31], gastric cancer[32] and T-cell lymphoma[33], indicating this miRNA act as a pancancer regulator with functions regardless of tumor origin.

We next inspected the regional distribution of CNV-miRs and DNAm-miRs in the human genome, surprisingly, we found that both groups of miRNAs are mainly distributed on chr14, followed by chr1 and chr19, which indicated the regional preference of miRNA expression that attribute to DNAm and CNV regulation (Fig. 1b). Furthermore, there is a significant overlap between the two miRNA sets (Fig. 1c), indicating that a large portion of miRNAs are coordinated regulated both by CNV and DNAm, whereas others are mutually exclusive regulated. Additionally, we also checked the positional distribution of the probes within miRNA genes and found that DNAm-miRs probes are more frequently located in promoter regions within 2000 bp upstream from the TSS to 200 bp upstream from the TSS (TSS2000) rather than within 200 bp upstream of the transcription start site (TSS200) and gene body (Fig. 1d, $p = 6.70e{-}43$, chi-square test). Moreover, DNAm-miRs probes are more frequently located in N-shores rather than in S-Shores, OpenSea and CpG island regions (Fig. 1e, $p = 6.14e{-}103$, chi-square test). We finally investigated the functions of the two different groups of miRNAs by performing the Gene Ontology enrichment analysis based on their target genes. To do this, we first performed multiple linear regression to dissect the DNAm- and CNV-miRNA expression associations. In this way, we identified a core subset of 425 DNAm-miRs and 286 CNV-miRs that uniquely influenced by these two different mechanisms. Enrichment analysis indicated these DNAm-miRs are mainly involved in nervous system development and brain morphogenesis as well as some tumor related pathways, such as Wnt signaling and Notch signaling (Fig. 1f). Whereas for CNV-miRs, functions are mainly about neuron development, cell proliferation and immune response pathways (Fig. 1g). These observations indicated that CNV and DNAm mediated miRNA dysregulation may fulfill diverse functions in LGG, and thus it could help to define patients as more appropriate subtypes and to find suitable treatment methods.

### Molecular subtypes defined by CNV-miRs and DNAm-miRs

Next, we investigated whether the expression of CNV-miRs and DNAm-miRs can be used for patients' stratification which associated with prognosis. We applied the NMF clustering analysis to miRNA expression with the cluster number $k$ set as 2–10, the $k$ values were determined for both CNV-miRs and DNAm-miRs, respectively. The optimal number of $k$ was selected as 4 based on expression profile of DNAm-miRs (Fig. 2a and Supplementary Fig. 6), and as 3 based on expression profile of CNV-miRs (Fig. 2b and Supplementary Fig. 7). Kaplan–Meier plot analyses demonstrated that the subtypes are significantly associated with survival difference for both clustering results identified from DNAm-miRs and CNV-miRs (Fig. 2c, d). In addition, we compared the clustering results and the histological types and

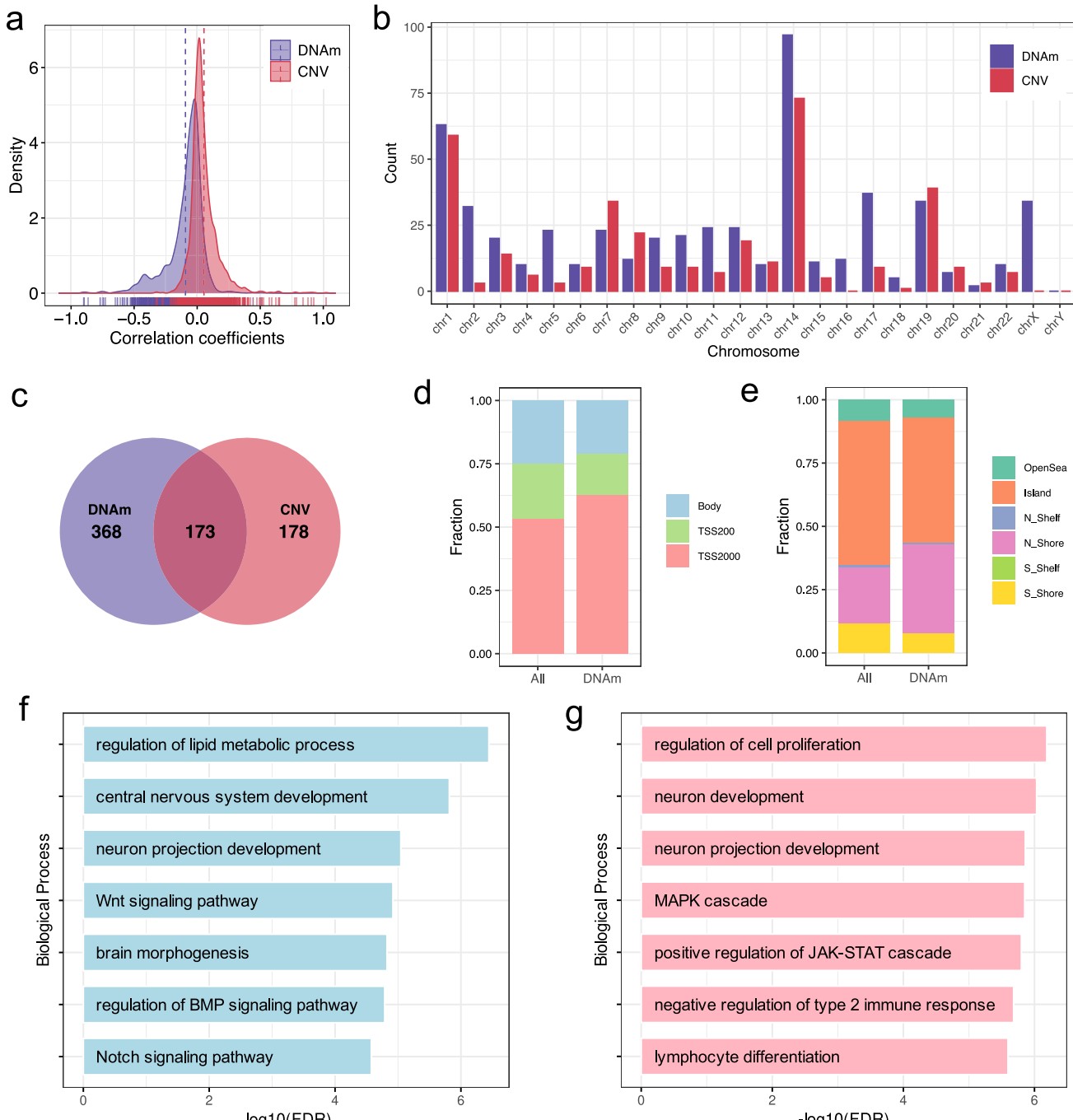

**Fig. 1 | Characteristics of the copy number related miRNAs (CNV-miRs) and DNA methylation related miRNAs (DNAm-miRs) in LGG. a** Distribution of Z-transformed correlation coefficients between miRNA expression level and copy number or DNA methylation across samples; **b** Frequencies of CNV-miRs and DNAm-miRs across different chromosomes; **c** Venn diagram shows the count of overlaps between CNV-miRs and DNAm-miRs; **d, e** The proportional frequencies in each category of DNA methylation probes for DNAm-miRs and all miRNAs, respectively. Methylation probes are categorized based on the positional relations with transcription start site and CpG islands, respectively; **f, g** GO terms for biological processes enriched by target genes of DNAm-miRs and CNV-miRs, respectively.

found that these classifications are distinct from each other (Fig. 2e), whereas the subtypes identified based on CNV-miRs overlapped to a large extent with those based on DNA-miRs (Fig. 2f), a large proportion of CNV_C1 cases also belonged to the DNAmC2 subtype. Such findings were consistent with the coordinated regulation of the CNV-miRs and DNA-miRs, and indicate the heterogeneous for LGG samples at molecular level.

We next sought to identify molecular subtypes that reflected the multi-layered pattern of the CNV-miRs and DNAm-miRs. We employed an integrated clustering method MOVICS for the combined analysis of the genome-wide profiling data regarding miRNA expression, DNAm and CNV.

We identified four integrative clusters (ICs) from the 10 multi-omics ensemble clustering algorithms, which were determined by referring to the cluster prediction index, gap statistical analysis and silhouette score, and was comprehensively evaluated by the consensus clustering matrix (Supplementary Fig. 8a–c). Then, the clustering results were further combined by the consensus ensemble approach with distinctive molecular patterns of the multi-omics data (Fig. 3a). Analysis indicated that our classification was closely associated with the survival of the patients (Log-rank $p = 3.15e-34$; Fig. 3b). Notably, integrative subtype 2 (IC2) exhibited the worst prognosis among all subtypes evaluated (Fig. 3c). For purpose of comparison, we also

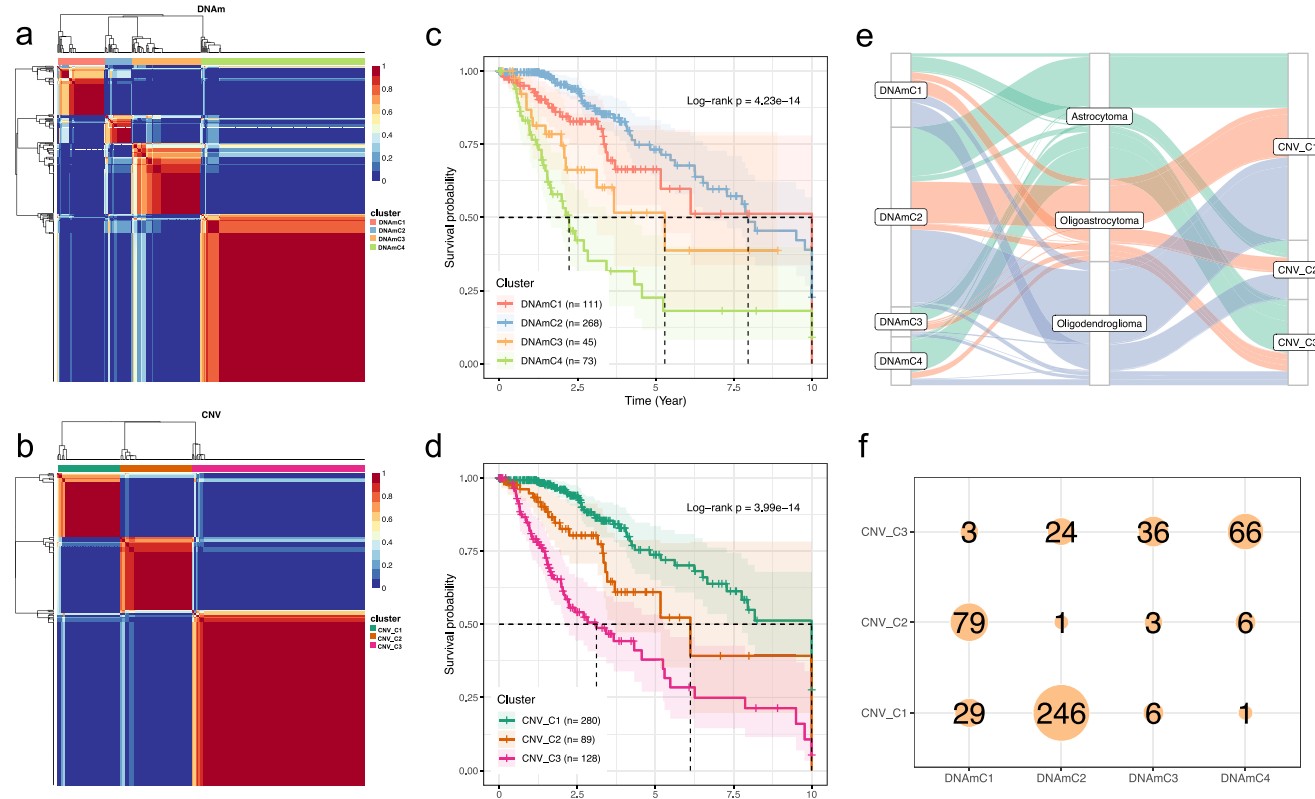

**Fig. 2 | Identification of LGG molecular subtypes based on expression profiling of CNV-miRs and DNAm-miRs. a** NMF clustering results for DNAm-miRs; **b** NMF clustering results for CNV-miRs. Kaplan–Meier plot indicated differences for OS among subtypes identified by NMF clustering of DNAm-miRs (**c**) and CNV-miRs (**d**), respectively; **e** Alluvial plot illustrates the overlap between the molecular subtypes identified from CNV-miRs and DNAm-miRs, as well as the histological subtypes; **f** Sample number of the overlaps between the CNV-miR and the DNAm-miR subtypes.

examined the overall survival among histological types, and only mild differences for their prognosis can be observed (Log-rank $p = 0.005$; Fig. 3d). Further examination to the sample distribution indicated there are higher proportions of overlapping between histological type of astrocytoma and IC1 cluster, and between oligodendroglioma and IC3 cluster (Fig. 3e). These results indicated that the integrative analysis of multi-omics data identified clinically-relevant molecular subtypes in which miRNA dysregulation caused by epigenomic and genomic aberrance influences prognostic outcomes.

We further compared clinical characteristics including the tumor grade, 1p/19q co-deletion, *MGMT* promoter methylation and *TERT* expression status among different clusters. We found that patients in these groups exhibited asymmetric distributions for these characteristics. For example, the major proportion of IC2 samples are grade III tumor, which is significantly higher than that of other groups ($p < 0.0001$; Supplementary Fig. 9a). In addition, all the samples within IC1 and IC2 cluster present 1p/19q co-deletion, in contrast, none of the samples in IC3 present co-deletion, and 64% of the samples in IC4 present co-deletion ($p < 0.0001$; Supplementary Fig. 9b). For the *MGMT* promoter methylation, IC2 group presents heights proportion ($p < 0.0001$; Supplementary Fig. 9c), but for *TERT* expression, IC2 and IC3 present relative lower proportions ($p < 0.0001$; Supplementary Fig. 9d). Finally, we evaluated the prognostic predictive value of the tumor subtypes identified in different grades. Stratified survival analyses showed that patients in the IC2 cluster have worse prognosis than patients in the other groups from both grade II and III samples (Supplementary Fig. 9e, f), which indicated the prognostic independence of our classifications method for patients' survival of LGG.

## Distinct genomic features among different molecular subtypes

Next, we examined the single nucleotide variation in different subtypes to inspect the molecular heterogeneity of LGG samples. We screened a set of

genes that altered significantly amongsubtypes obtained, and surprisingly, the somatic mutational spectrum of these genes agrees fantastically with the subtypes identified. As shown in Fig. 4a, we found that *IDH1* mutates predominantly in IC1, IC3 and IC4, and the mutation frequency of this gene presents significant differences (chi-square test $p = 3.53e-64$, Fig. 4b), whereas the mutation of *TP53* predominates in IC1 (chi-square test $p = 1.85e-66$, Fig. 4b). In contract, IC2 cluster bears nearly all the mutations for *EGFR* (chi-square test $p = 3.33e-29$, Fig. 4b). A mutually exclusive mutation pattern for the *IDH1* and *EGFR* can be observed. In addition, some other mutant genes including *ATRX*, *CIC* and *FUBP1* were also identified. We next checked the associations between mutational spectrums of these genes and the patients' survival. In general, only mild or no survival differences can be observed between groups defined by the mutational status of these genes (Supplementary Fig. 10). This analysis indicates that our suggested classification system is superior to that based on mutations of individual genes for patients' prognosis of LGG. We further extended our analysis to all mutated genes and calculated tumor mutation burden (TMB) for the samples, as a result, IC2 presents highest mutation load across all clusters (Fig. 4c), which indicated genetic heterogeneity are highly correlated with the subtypes defined by DNAm and CNV related miRNAs.

Inspired by the correlations between the mutational spectrum and the multi-omics clustering results for LGG, we further checked the pathways affected in order to obtain the potential targets for precise treatment, particularly for IC2. By using maftools, we identified the most significant pathways including "RTK-RAS", "WNT", "NOTCH", etc., which are differentially affected between IC2 and other groups for the genes involved (Fig. 4d). The potential druggable gene categories for IC2 and other groups are shown in Fig. 4e, f. We could see that the top list genes for IC2 includes the *COL6A3*, *EGFR*, *MUC16*, *OBSCN* and *PIK3CA*, whereas *ARID1A*, *ATRX*, *BCOR*, *CIC* and *FUBP1* are included for other groups. These

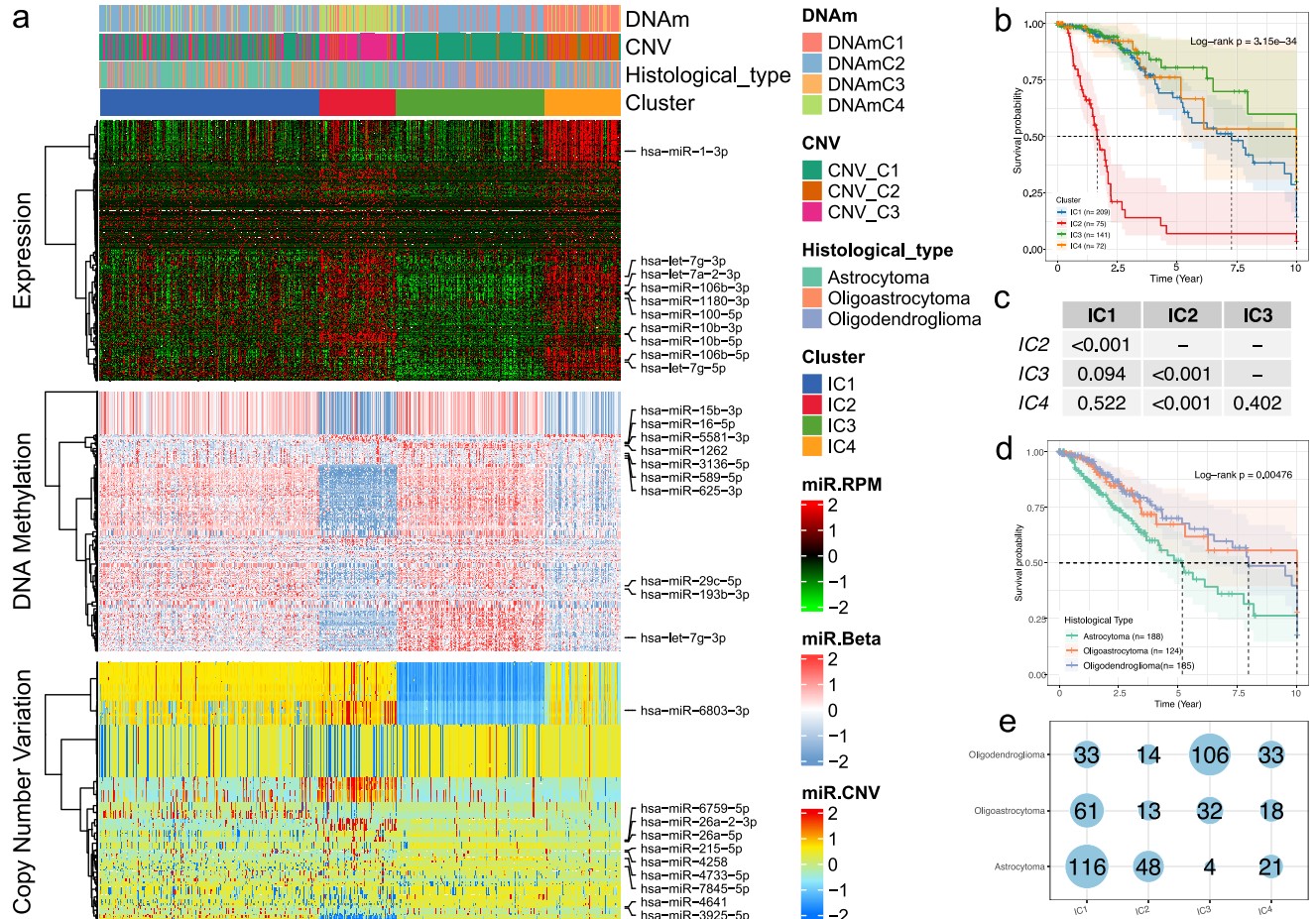

**Fig. 3 | LGG sample classification by integrative analysis of multi-omics data.**
**a** Comprehensive clustering heatmaps for miRNA expression, DNA methylation and copy number variation identify consensus ensemble subtypes of LGG. Samples are labeled for the subtypes identified by CNV-miR, DNAm-miR and also the histological subtypes; **b** Kaplan–Meier plot indicated differences for survival among ensemble subtypes; **c** *p* values generated by log-rank test that among different integrative clusters; **d** Kaplan–Meier plot indicated differences for survival among histological subtypes; **e** Sample number of the overlaps between the integrative clusters and the histological subtypes.

intriguing results strongly suggest that the distinct mutational events might synergistically involve in miRNA based LGG subtyping and thus provides different insights to development of potential therapeutic strategies for clinical utility.

**Immune landscape of LGG molecular subtypes**

The composition of the tumor immune microenvironment (TIME) has been identified as the critical factor of tumor–immune interactions to influence clinical outcome of patients and also the response to immune treatment[34]. To evaluate whether the difference of the multi-omics pattern imply the TIME heterogeneity of LGG, we investigated immune cell infiltrations and immune functions discrepancy among different subtypes. We first focus on 17 immunologically relevant pathways derived from ImmPort[35]. Gene expression fold changes between IC2 and other clusters were calculated and GSEA was performed for the collected immune pathways. Analysis results showed that more than half of the pathways are enriched by the genes present dysregulation, which include the interleukins, chemokines, cytokines and cytokine receptors (Fig. 5a). Whereas for other clusters, functions are mainly enriched in BCR and TCR signaling, antigen processing and presentation, and cell cytotoxicity related pathways (Supplementary Fig. 11). Next, we calculated the immune score by ESTIMATE for each tumor sample to evaluate the immune infiltration degree[36]. We observe the immune scores were significantly increased in IC2 cluster, which indicated patients in IC2 cluster present higher infiltration of immune cells in tumor samples (Fig. 5b). In addition, IC2 cluster also yields higher stromal

scores than other groups, but lower tumor purity scores obtained by ESTIMATE (Wilcoxon *p* < 0.05, Supplementary Fig. 12), which indicate that IC2 cluster samples are also more heterogeneous than that of other groups. Moreover, we also calculated the antitumor immunoactivity for glioma patients, including CYT, MHC and CTL scores. All the three immunoactivity scores were significantly increased in IC2 cluster (Fig. 5c), indicating patients in IC2 cluster displayed greater tumor cell cytotoxic activity and stronger antigen recognition capacities. In addition, we collected eight different immune signature gene sets and the ssGSEA was uses to estimate the enrichment score of tumor samples (Supplementary Data 5), also, IC2 cluster presents higher enrichment of all immune signatures (Fig. 5c). For instance, higher IFN-γ score in IC2 cluster indicated that IC2 have stronger antitumor immunity and inhibited angiogenesis activities in tumor, thus to influence the efficiency of cancer immunotherapies[37].

We next evaluated the relative proportion of immune cells of tumor samples using TIMER. Similarly, we found the infiltrate levels of CD4+ T cell, neutrophil, macrophage and dendritic cells were highest in IC2 among all clusters (Fig. 5d), which presents the typical phenotypes of immune "hot" tumor for IC2. Finally, we collected 63 well-known marker genes for 21 immune cell types from public resources (Supplementary Data 6), the clustering heatmap shows that these genes exhibited diverse expression patterns among clusters, where samples in IC2 showed higher expression in general. (Fig. 5e). Collectively, these results suggested the DNAm and CNV mediated miRNA expression heterogeneity is highly correlated with tumor immune status and could determine the immunophenotypes of glioma.

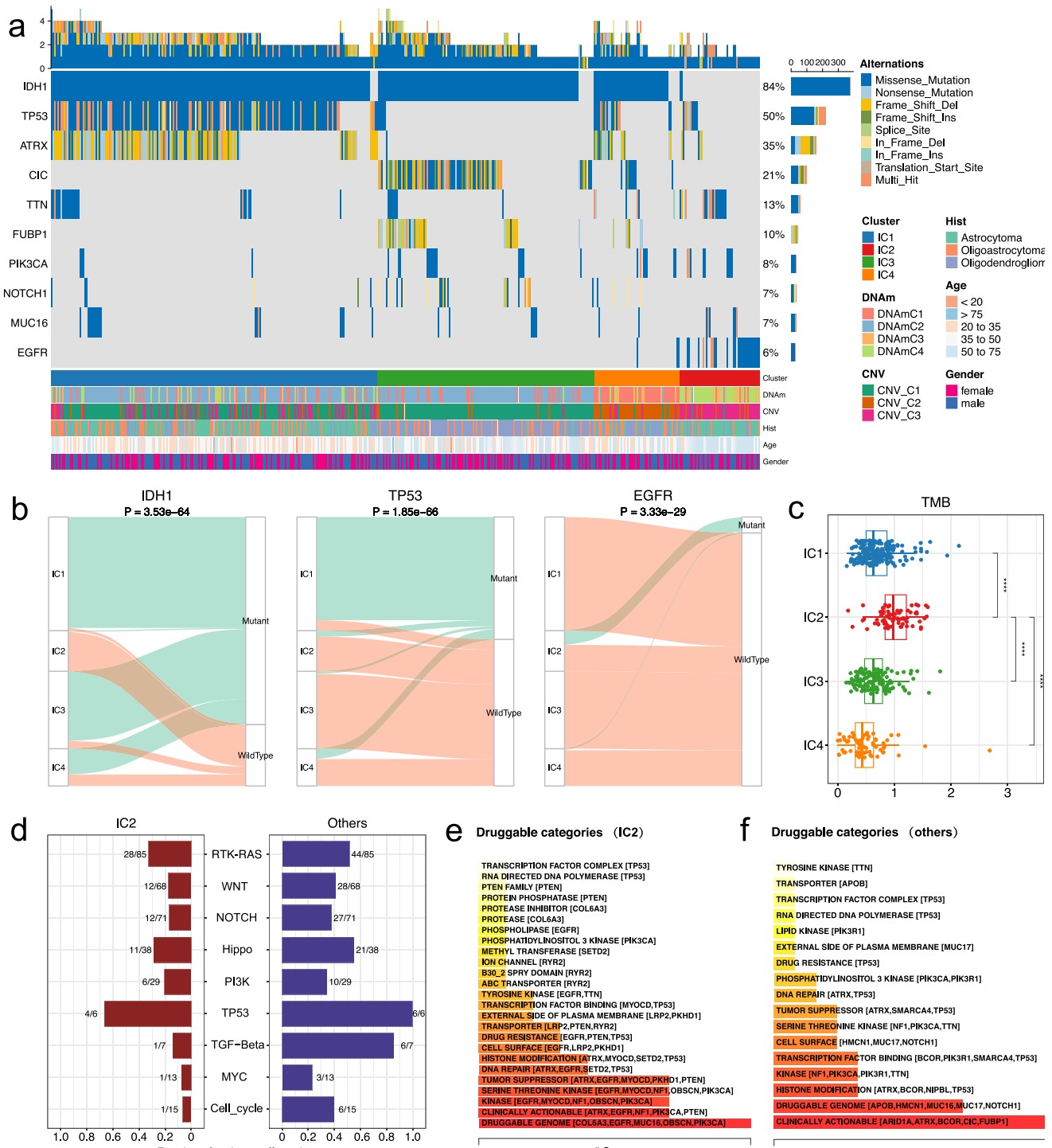

**Fig. 4 | The landscape of genomic features of LGG subtypes. a** Oncoplot showing the differentially mutated genes among the subtypes of LGG; **b** Alluvial plot illustrates the overlap between the LGG subtypes and mutational status of *IDH1*, *TP53* and *EGFR*, respectively; **c** Boxplot showing the distribution of tumor mutation burden among different clusters; **d** Bar plot of the pathway alteration frequency and the fraction of sample affected for IC2 and other clusters; **e** Potentially druggable gene categories from mutation datasets for IC2; **f** Potentially druggable gene categories from mutation datasets for other clusters.

## Immune phenotype associated miRNA identification and prognosis model construction

Having demonstrated the relationship between immune phenotypes and the DNAm as well as CNV associated miRNA expression dysregulation, we next sought to comprehensively identify the miRNAs that determine the immune cell infiltrations and immune functions. By calculating the Pearson correlation coefficient between cell fractions and miRNA expression, we identified 517 DNAm-miRs and 325 CNV-miRs that correlated with infiltrating immune cells, respectively (Fig. 6a). In general, the number of positively correlated miRNAs is higher than that of negatively correlated ones for both groups. Interestingly, we observe that the number of miRNAs with immune cell infiltration associations increases with that of cell types, and there are 195 miRNAs were identified that correlated with infiltrations of all six cell types (Fig. 6b and Supplementary Data 7). In addition, we observe that miRNA expression correlated with diverse immune cell type infiltration in majority of the clusters identified. Most miRNAs are positively

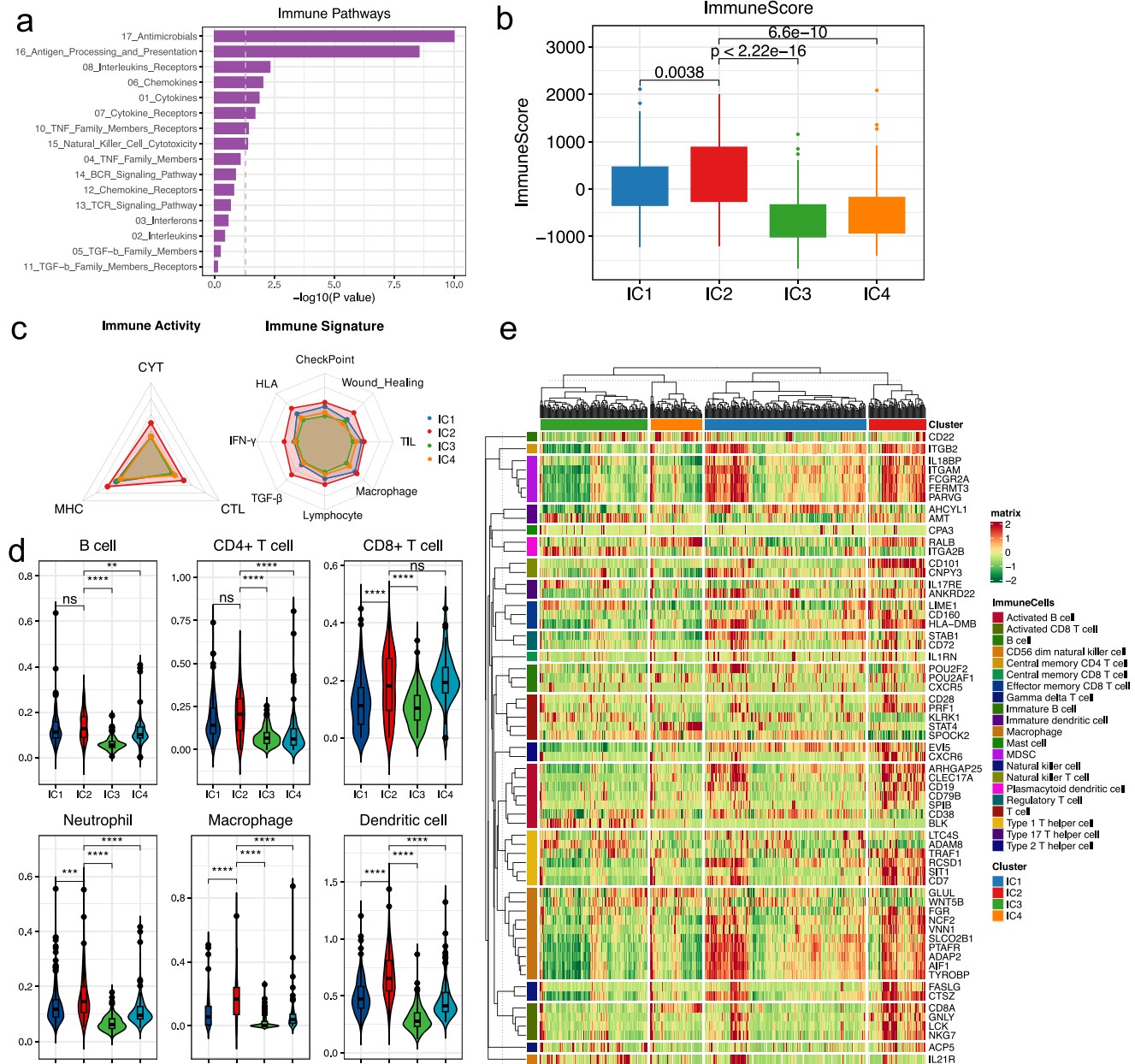

**Fig. 5 | Characteristics of tumor immunophenotypes in LGG subgroups. a** Bar plot showing immune-related pathways enriched by genes differentially expressed between IC2 and other clusters, significant threshold at *p* < 0.05 are indicated; **b** Boxplot showing the distribution of the ESTIMATE immune score across different clusters; **c** Radar plot showing the mean value of scaled antitumor immunoactivity and immune signature scores across different clusters; **d** Violin plots showing the immune cell infiltration levels obtained from TIMER in different clusters; **e** Heatmap showing the relative expression of immune cell type marker genes in different clusters.

associated with immune cells, whereas the negative correlations are mainly occurring in CD4+ T cell (Fig. 6c). In order to further explore key miRNAs that involved in immune regulation for LGG, we collected experimentally verified immune-related miRNAs from the RNA2Immune database, which is a public repository to catalog ncRNA-immune associations[38]. We selected 464 and 574 miRNAs from the sections of "immune cell function" and "cancer immunity", respectively (Supplementary Data 8). By comparing these miRNAs with those correlated with immune cell infiltrations, a total of 64 miRNAs were identified as the core set for immune regulation (Fig. 6d), in which 47 miRNAs are from DNAm-miRs and 17 are from CNV-miRs (Fig. 6e).

Studies have extensively indicated that miRNAs exhibited strong potential for predicting cancer prognosis and for clinical applications as therapeutic targets[39]. Thus, we integrated the survival information to

evaluate the potential ability of the identified miRNA as candidate prognostic marker for LGG. The Cox-proportional hazard regression analyses was used to identify the survival related miRNAs. Under a stringent threshold of *p* < 0.0005, a total of 34 miRNAs were identified that correlated with patients' survival. We found that almost all of these miRNAs are positively correlated with immune cell infiltration levels, and only miR-9-5p exhibits negative correlations (Fig. 6f). We further performed LASSO proportional hazards regression analysis to determine the relationship between the expressions of the 34 miRNAs and patients' survival. We found a strong correlation for the overall survival of LGG patients with the following 8 miRNAs: (hsa-miR-10b-5p, hsa-miR-155-5p, hsa-miR-196a-5p, hsa-miR-196b-5p, hsa-miR-200a-3p, hsa-miR-204-5p, hsa-miR-503-5p, hsa-miR-15b-5p). These miRNAs were considered to function as prognostically significant immune-related biomarkers. We then implemented these 8

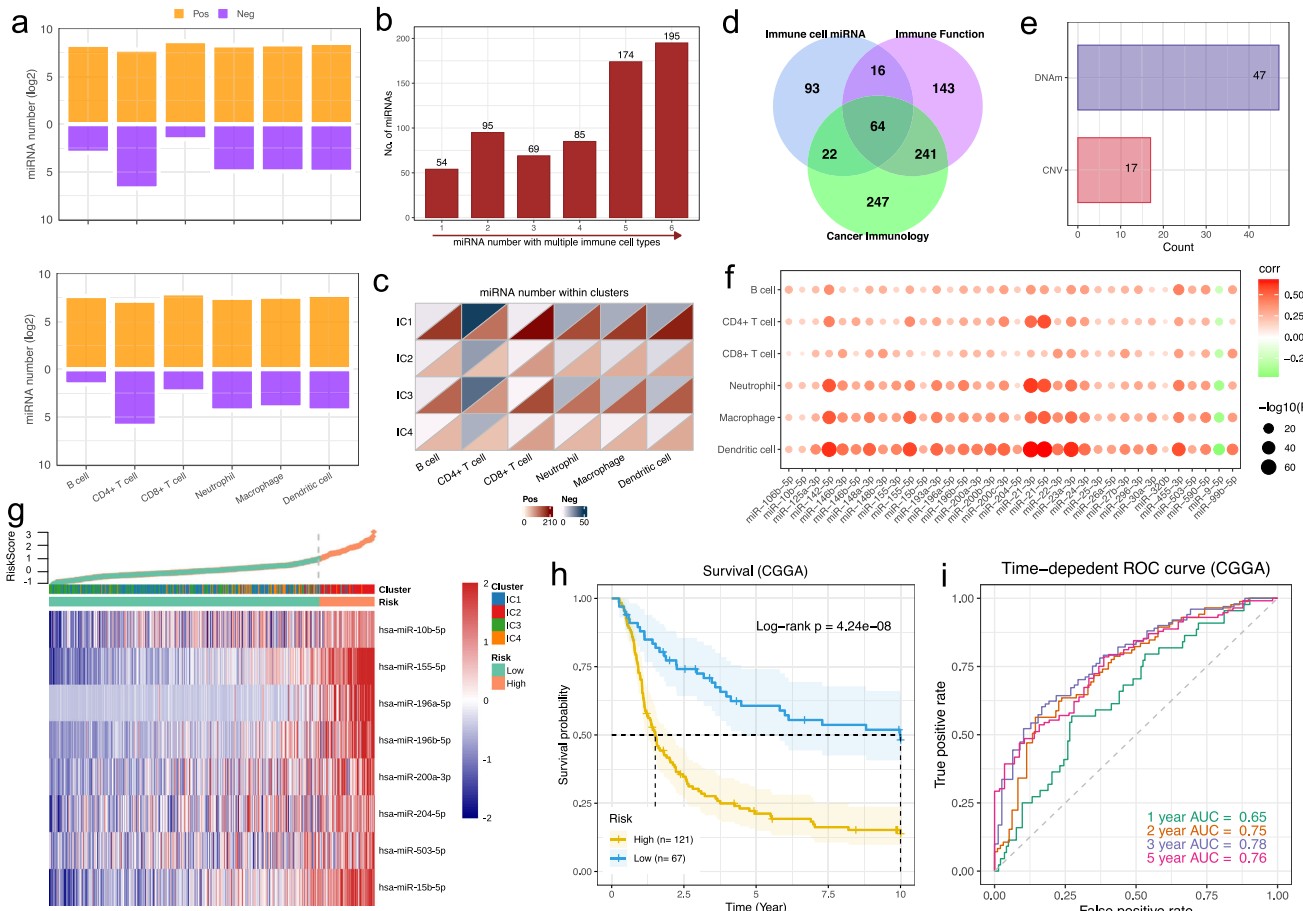

**Fig. 6 | miRNAs correlated with immune cell infiltrations in LGG. a** Bar plots showing the number of DNAm-miRs and CNV-miRs in which the expression correlated with immune cell infiltrations, respectively. The number of positive and negative correlated miRNAs are indicated by yellow and purple, respectively; **b** The number of miRNAs that correlated with different numbers of immune cell types; **c** Heatmaps showing the number of miRNAs in which expression correlated with different types of immune cell infiltrations within different clusters; **d** Venn diagram indicated the number of specific and shared miRNAs that related to immune cell and that from sections of "Immune function" and "Cancer immunology" of RNA2Immune; **e** Bar plots shows the number of immune miRNAs from DNAm-miR and CNV-miR, respectively; **f** Bubble plots of correlation between expression and immune cell infiltration for prognosis related miRNAs; **g** Risk score and expression heatmap of the eight signature miRNAs in the training set; **h** Kaplan–Meier plot of survival for patients with different risk scores from an independent validation cohort of CGGA; **i** Time dependent ROC curve and AUC indicated the prognostic performance for survival prediction in validation set.

miRNAs in development of a prognostic risk score by weighting the normalized expression level of each immune miRNA to the LASSO regression coefficient. We calculated the risk score for each patient and observed that the IC2 cluster presents height risk scores among all groups as expected (Wilcoxon $p < 2.2\mathrm{e}{-}16$, Supplementary Fig. 13). The risk score distribution and miRNA expression are indicated in Fig. 6g. All the samples in this training set were categorized into high-risk ($N = 84$) and low-risk groups ($N = 411$) according to the risk scores, we can observe a clear distinction between for the signature miRNA expression from the two groups of patients.

In order to determine whether the prognostic model was robust, a validation set obtained from CGGA was utilized to verify the validity and accuracy of the prognostic model based on the eight miRNAs. We collected miRNA microarray data for 198 samples from this cohort, 188 of which have the survival data available, each sample of the validation set also acquired a risk score according to the same formula, and was then stratified as being either high-risk ($N = 121$) or low-risk groups ($N = 67$). The risk score distribution and gene expression data for the validation set are shown in Supplementary Fig. 14. Consistent with the outcomes of the TCGA cohort, patients with higher risk scores were noted to have worse survival in contrast to those with lower risk scores (Log−rank $p = 4.24\mathrm{e}{-}08$, Fig. 6h). The predictive potential of the prognostic model using time-dependent

ROC curves was shows as Fig. 6i. The area under the ROC curve (AUC) of the prognostic model for OS was 0.65 at 1 year, 0.75 at 2 years, 0.78 at 3 years and 0.76 at 5 years, suggesting that our model has favorable efficacy for predicting both short- and long-term prognosis.

## qRT-PCR validation of signature miRNAs and clinical utility of risk model

We next validated the miRNAs as prognosis markers for glioma in our in-house cohort. We detected the expression level of the eight miRNAs form tumor and normal samples of 30 LGG patients by qRT-PCR (Supplementary Data 9). The normalized expression levels from qRT-PCR showed that all these miRNAs present elevated expression in LGG tumor sample. Two miRNAs of miR-10b-5p and miR-204-5p present mild upregulation for about 1.5 times of fold change, whereas others are significantly upregulated in about 20–50 times, Notably, the miR-155-5p showed highest level of fold change in 56.4 times (Fig. 7a). These cross platform and cross-racial data indicated the robustness of these signature miRNAs presented here for patient's prognosis.

Furthermore, we conducted univariate and multivariate Cox regression analyses to explore whether the prognostic value of our risk model was independent of other clinical features in the TCGA cohort. For the univariate Cox regression model, our prediction classifier was the strongest

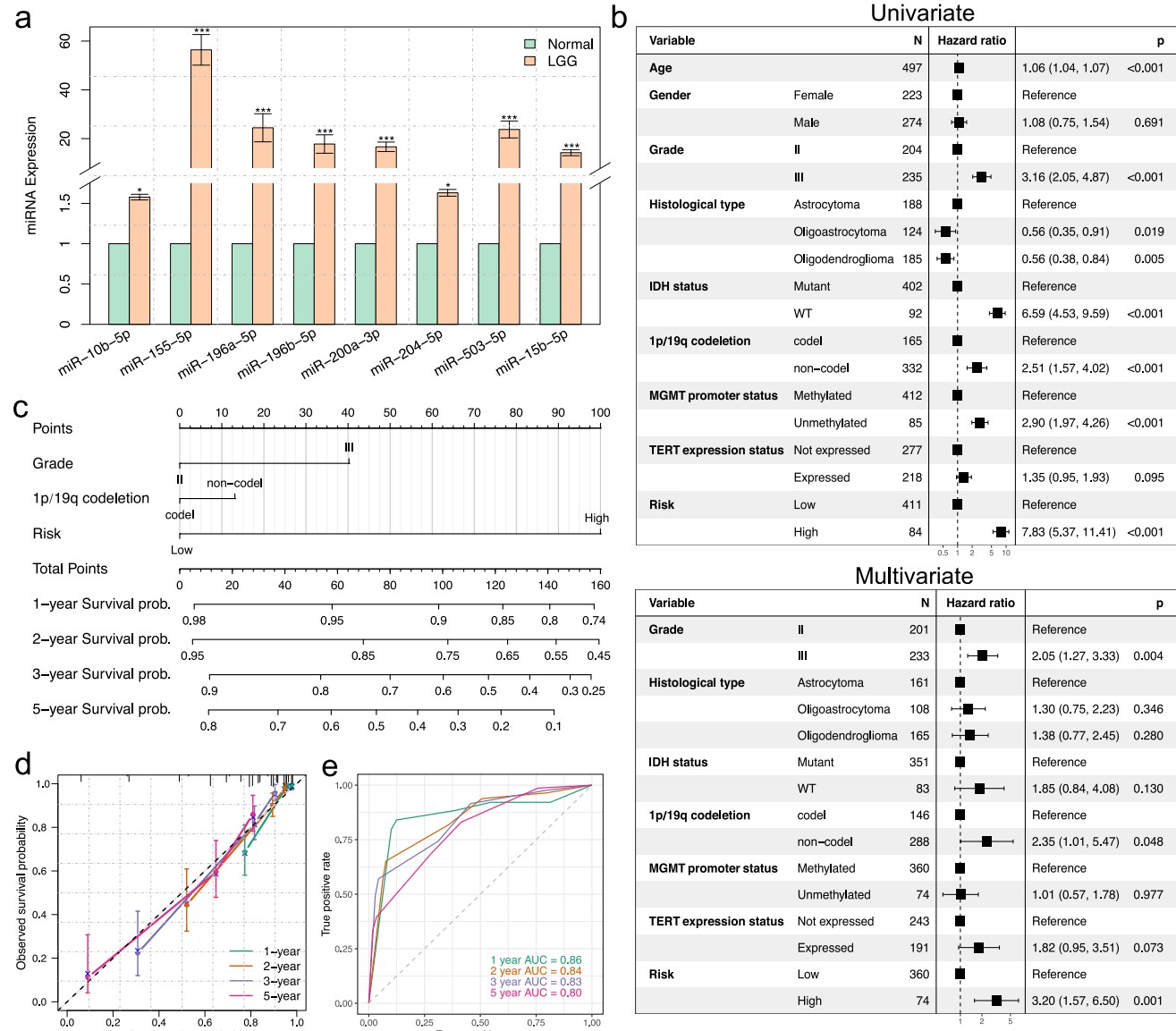

**Fig. 7 | Relationship between the risk model and other clinical information.**
**a** qRT-PCR validation for the expression of eight signature miRNAs of our in-house cohort. The results are shown as the mean ± SEM of 30 biologically independent samples (*$p < 0.05$, ***$p < 0.05$ and fold >2); **b** Univariate and multivariate regression analysis of the relation between our prognostic model and other clinicopathological features regarding prognostic value; **c** Nomogram for predicting the probability of 1-, 2-, 3- and 5-year survival for LGG patients; **d** Calibration plot of the nomogram for predicting the probability of patient's survival at 1-, 2-, 3- and 5-years; **e** Time-dependent ROC curves based on the nomogram for 1-, 2-, 3- and 5-year survival probability.

variable correlated with prognosis among clinical features including histological type, *IDH* mutation, 1p/19q codeletion, *MGMT* promoter methylation and *TERT* expression status. After multivariate adjustment by the clinical features, our classifier remains the strongest and independent prognostic factor for survival analysis (Fig. 7b). This results indicated that our model performed better than conventional clinicopathological features for clinical outcome prediction. To provide clinically correlated quantitative approach for predicting the prognosis of LGG patients, a nomogram that integrated the risk model and independent clinical risk factors (tumor grade and 1p/19q codeletion) was constructed (Fig. 7c). Our risk model was found to contribute the most risk points (ranging from 0 to 100) compared with the other clinical features, which was consistent with the Cox multivariate regression results. The calibration plot indicated that the nomogram performed well compared with an ideal model (Fig. 7d). The AUC score at 1-, 2-, 3- and 5-years were 0.86, 0.84, 0.83 and 0.80 for the nomogram, respectively (Fig. 7e). In summary, these findings suggest that the nomogram is a better model for predicting short-term or long-term survival of LGG patients than individual prognostic factors.

## miRNAs associated with immune evasion of LGG

Since the IC2 group exhibited higher levels of immune cell infiltration, whereas patients in this cluster have poorer prognosis than others, we conducted in-depth functional investigations to explore possible reasons for this contradiction. We performed functional enrichment analysis of GSVA on Hallmark gene sets obtained from MSigDB[40]. Specially, the IC2 group is endowed with higher energy expenditure states, such as glycolysis, hypoxia and angiogenesis, whereas IC4 is more closely related to metabolic activities, such as the bile acid and fatty acid metabolism, as well as protein secretion. In contract, IC3 and IC1 groups are more distinguished by proliferative-related pathways, such as mitotic spindle, E2F targets, and G2M checkpoints, and also by classical tumor pathways including the MYC and Wnt-signaling (Fig. 8a). Next, we investigated whether the clusters obtained are

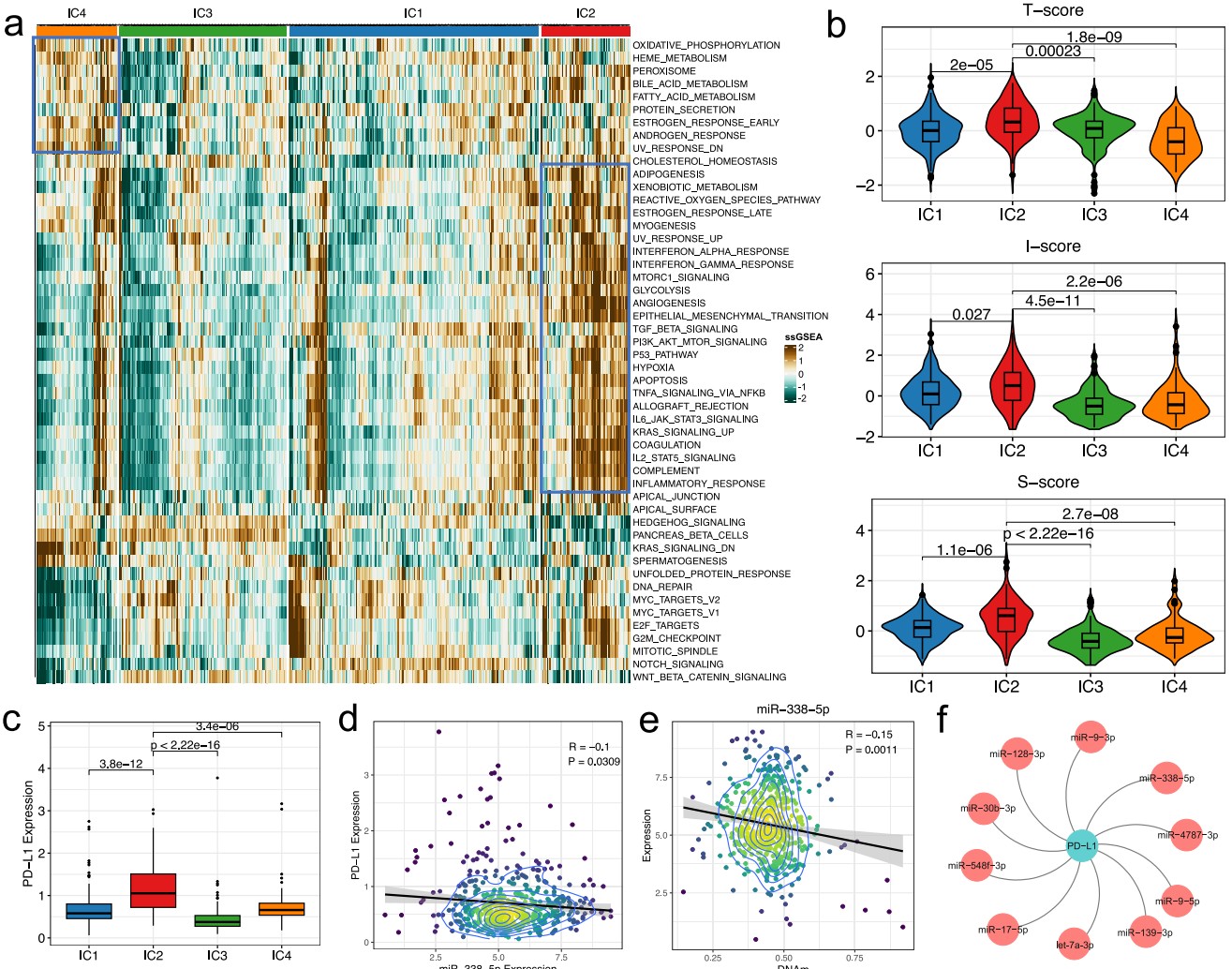

**Fig. 8 | Biological function and immune evasion of LGG subtypes. a** Heatmap of the enrichment score obtained from ssGSEA on Hallmark genesets; **b** Violin plots showing the distribution of "T-score", "I-score", and "S-score" among different clusters; **c** Boxplot showing the distribution of PD-L1 expression among different clusters; **d** Scatter plot showing significant correlation between expression of PD-L1 and miR-338-5p; **e** Scatter plot showing significant correlation between expression and promoter DNA methylation of miR-338-5p; **f** Regulatory module consists of PD-L1 and the associated miRNA regulators.

correlated with immunosuppressive effects. We use the methodology from Shuichi et al. to calculate three different scores in order to measure tumor proliferation and glycolysis rates (*T*-score), immune cell and antitumor immunity level (I-score), as well as immunosuppressive level (S-score), respectively[41]. Here, the IC2 cluster conveys the highest for all three different scores, particularly for the S-score (Fig. 8b), which indicated an immune evasion state may occurred in the patients of IC2 cluster. Next, we inspected *PD-L1* expression among samples and found its expression is much higher in IC2 than other groups (Fig. 8c), which verify the hypothesis of an immune evasion status for the IC2 cluster.

To further identify key miRNAs that involved in LGG immune evasion, we collected experimentally validated regulatory interactions between *PD-L1* and miRNAs from miRTarBase and further screened those interactions by performing a Pearson correlation analysis between the expression of *PD-L1* and DNAm-miRs as well as CNV-miRs. Based on negative correlation of PCC < 0 and the threshold of FDR < 0.05, we identified miRNAs that play key roles in immune evasion possibly by regulating *PD-L1* expression. One of the examples is the miR-338-5p, our analysis indicated this miRNA presents significant negative correlation with *PD-L1* for their expressions (*r* = −0.1, *p* = 0.0309) (Fig. 8d), and a negative correlation between expression of this miRNA and promoter methylation was observed (*r* = −0.15, *p* = 0.0011) (Fig. 8e). This regulatory relationship has been

identified for regulatory T cells (Tregs) mediated immune evasion[42]. Another example is the miR-17-5p, whose expression is significantly negatively correlated with that of *PD-L1* (*r* = −0.12, *p* = 0.0056) (Supplementary Fig. 15a), its expression dysregulation was identified that caused by the copy number variation (*r* = 0.24, *p* = 7.14e−08) (Supplementary Fig. 15b). Such inverse correlation between *PD-L1* and miR-17-5p has been confirmed in many cancer types. For instance, up-regulation of *PD-L1* due to post-transcriptional control by miR-17-5p confers resistance to BRAF or MEK inhibitors (BRAFi or MEKi) for metastatic melanoma[43]. In addition, ceRNA regulatory loops involving miR-17-5p/PD-L1 have been observed in lung cancer and breast cancer to promote tumor progression and immune checkpoint blockades (ICB) resistance[44,45]. Other similar regulatory interactions were also identified for miRNAs including let-7a-3p, miR-30b-3p, miR-9-3p, miR-548f-3p, miR-139-3p miR-9-5p, miR-128-3p and miR-4797-3p. These miRNAs present significant negative correlations for their expression with *PD-L1* (Supplementary Fig. 16), which indicated this miRNA regulatory module may play critical roles in *PD-L1* associated immune evasion for LGG (Fig. 8f). Furthermore, we also examined the expression of several other key genes associated with immune regulation in different groups, including *KLRB1*, *CD70*, and *FASLG*, which have been shown to play key roles in glioma immune evasion[46–48]. Results indicated that all these genes present significantly elevated expression in IC2 group,

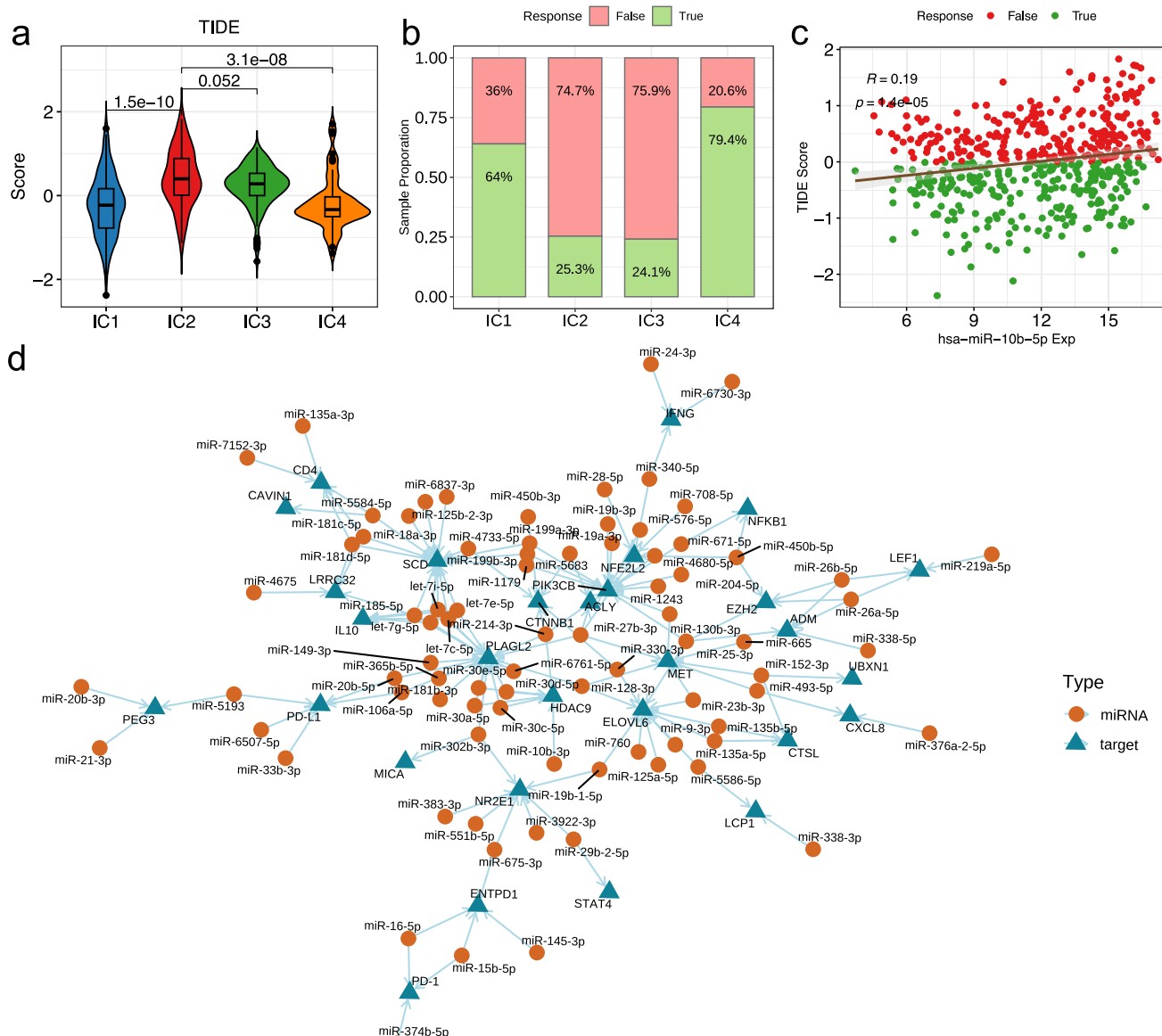

**Fig. 9 | Identification of miRNA immunoevasive biomarkers. a** Violin plots showing the distribution of TIDE score among different clusters; **b** Stacked bar chart of the fractions of immune therapy response for patients among different clusters; **c** Scatter plot showing significant correlation between TIDE score and miR-10b-5p expression; **d** miRNA immune evasive network for the putative targets and immune evasive miRNAs in LGG.

their associated miRNA regulatory modules were also identified (Supplementary Fig. 17a–c), which confirmed miRNA related gene expression regulation are widely involved in immune evasion for LGG.

To further validate our observations on the immunoregulatory status of glioma, we then used single-cell transcriptome data to analyze cell compositions and immune evasion related gene expression in different LGG samples. Since the IC2 group is more concentrated in *IDH* wild-type patients, while the other groups are more concentrated in *IDH* mutant patients, we used the single cell sequencing data associated with IDH status for validation purpose. We obtained single cell expression profiles based on 10x Genomics platform from 18 samples with either *IDH1* wild-type or mutant subtypes from GEO database with accession number GSE152273[49]. A total of 18,260 cells were retained for further analysis after quality control and standardization. UMAP plot revealed no noticeable batch effect among the samples (Supplementary Fig. 18a). Based on the expression of cells, a total of 17 cell clusters were obtained and annotated as astrocyte, epithelial cell, endothelial cell, T cell, macrophage, monocyte and tissue stem cells (Supplementary Fig. 18b). We can observe a higher proportion of immune

cells in *IDH1* wild-type samples, the most significant differences between *IDH1* mutant and wild-type samples is the T cell (Supplementary Fig. 18c). We next obtained the pseudobulk expression data by aggregating cells from each sample and then compared gene expression of the four immune genes of *PD-L1*, *KLRB1*, *CD70*, and *FASLG*. Among them, *KLRB1*, *CD70*, and *FASLG* are found that have significant higher expression level in *IDH1* wild-type samples (Supplementary Fig. 18d). These findings suggest that the heterogeneous pathogenesis of LGG may be associated with dysregulation of immune evasion related genes.

Finally, to provide more comprehensive view of miRNA related tumor immune evasion mechanism, we used TIDE method to evaluate the efficacy of ICB based immunotherapy for LGG[50]. We found patients in the IC2 and IC3 clusters presents significant higher TIDE scores than other groups, which results in a lower efficacy of immunotherapy (Fig. 9a). According to TIDE analysis, only about one quarter of the patients in IC2 and IC3 may response to *PD-1* and *CTLA-4* based treatment, in contract, about 64–79% of the patients in IC1 and IC4 could benefit from the treatment (Fig. 9b). By Pearson correlation analysis between TIDE scores and miRNA expression,

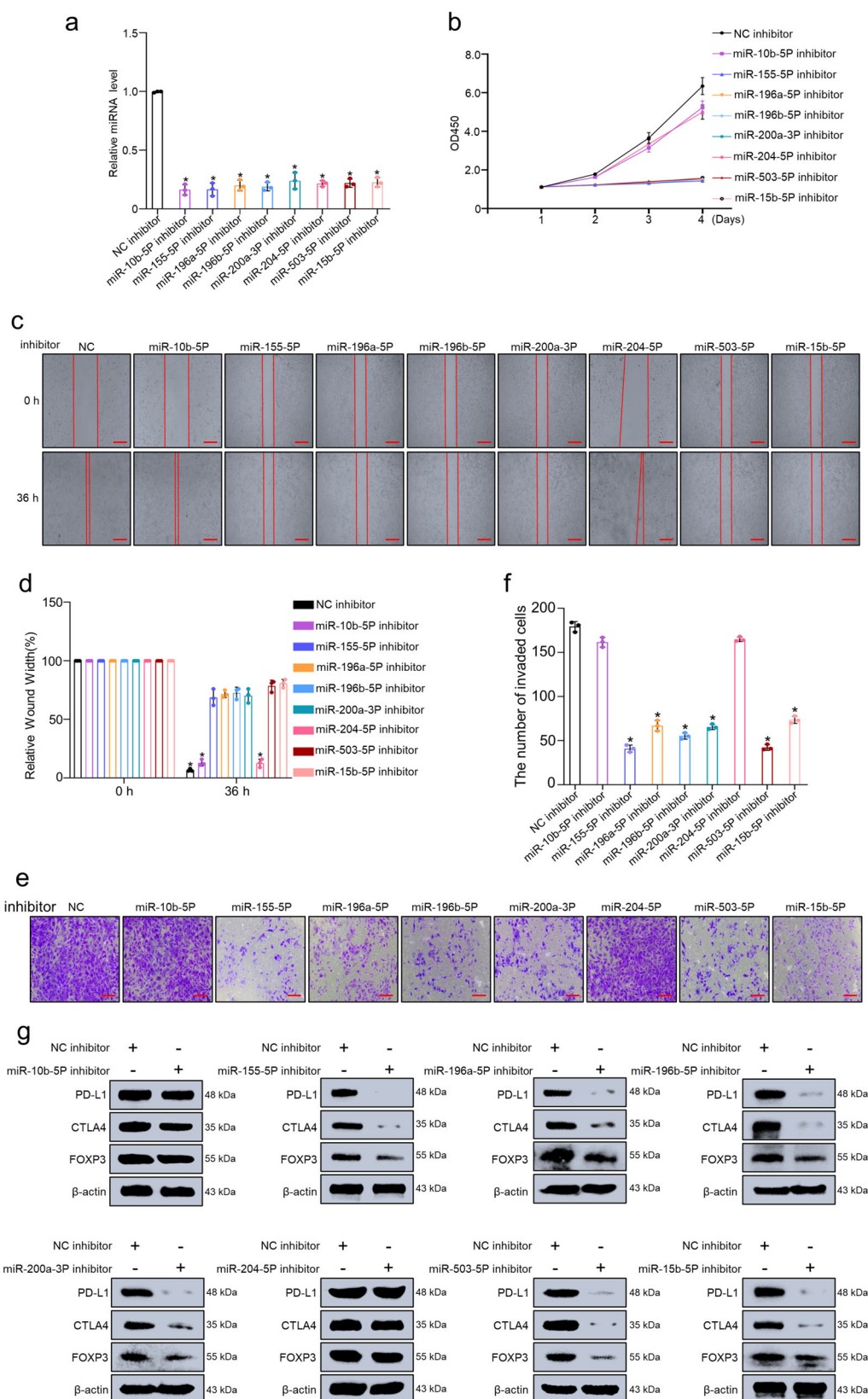

we identified 225 miRNAs with significant correlation under FDR < 0.05, which may involve in immunotherapy response, such as miR-10b-5p, a significant positive correlation was observed between TIDE score and miR-10b-5p expression (Fig. 9c). This miRNA has been identified that impairs TET2-mediated *PD-L1* transcription inhibition, by which to promote immune evasion and tumor progression in glioblastoma[51]. By further integration of the TIDE score correlated miRNAs and the putative targets obtained from public resources, we constructed a miRNA immune evasive network to describe the mechanism explaining roles of miRNA involved in immune envision regulation (Fig. 9d). Within this network, 29 unique

**Fig. 10 | Role of the prognosis related miRNAs in the proliferation, migration, invasion and immune evasion of glioma cell lines. a** The expression of miRNAs (miR-10b-5p, miR-155-5p, miR-196a-5p, miR-196b-5p, miR-200a-3p, miR-204-5p, miR-503-5p, and miR-15b-5p) in U251 cells was determined by qRT-PCR; **b** CCK-8 assays show that the inhibition of miRNAs (miR-155-5p, miR-196a-5p, miR-196b-5p, miR-200a-3p, miR-503-5p, and miR-15b-5p) decreased cell proliferation in U251 cell lines cells; **c, d** Wound healing assays show that miRNAs (miR-155-5p, miR-196a-5p, miR-196b-5p, miR-200a-3p, miR-503-5p, and miR-15b-5p) knockdown significantly reduced the cell migration ability of U251 cells

with the representative images and the quantitative analysis. Scale bars, 200 μm (×100 magnification). **e, f** Transwell invasion of miRNAs (miR-155-5p, miR-196a-5p, miR-196b-5p, miR-200a-3p, miR-503-5p, and miR-15b-5p) knockdown cells is significantly reduced compared with control cells. Scale bars, 100 μm (×200 magnification). **g** The expression of *FOXP3*, *CTLA-4* and *PD-L1* in U251 cells transduced with miRNAs (miR-10b-5p, miR-155-5p, miR-196a-5p, miR-196b-5p, miR-200a-3p, miR-204-5p, miR-503-5p, and miR-15b-5p) inhibitor was examined by Western blot; β-actin was an internal control. Data are the mean ± SD from three independent experiments. *$p < 0.05$.

targets of 82 immune envision related miRNAs that under the CNV and DNAm burden were identified. We screened genes that simultaneously targeted by two or more miRNAs and found *PLAGL2*, *SCD1*, *PIK3CB*, *ELOVL6*, *NR2E1* and *MET* demonstrated the highest connectivity, which suggests the importance of these genes in immune envision and the associated immunotherapy response for LGG. Actually, many of these genes have been validated for their functions in immune regulation and cancer development. For instance, the *PLAGL2* and the *PIK3CB* were identified as key hub gene within the regulatory network that significantly associated with tumor immune infiltration and sensitivity of chemotherapeutic drugs for cancer[52,53]. *SCD1* was found that expressed in cancer cells and immune cells causes immune-resistant conditions, inhibition of *SCD1* enhances the recruitment of dendritic cells into tumors and the subsequent induction and tumor accumulation of antitumor CD8+ T cells, thus this gene was identified as a potential target to enhance the antitumor effects of an anti-PD-1 therapy[54]. For miRNAs, miR-214-3p, miR-25-3p and miR-27b-3p were identified as key hub regulators with highest connectivity, and also have been validated for their roles in immune escape[55,56]. In summary, our analysis revealed miRNAs act as key regulators for genes that involved in cancer immune evasion, and thus could act as an adjuvant to several major immunotherapies by synergistically increasing their efficacy potential.

## Validation of the association between prognosis related miRNAs and immune evasion in vitro experiments

Finally, we evaluated the capacities of prognosis related miRNAs as immune evasion regulators for glioma in vitro experiments. miRNA inhibitor technology was used to knockdown the miRNAs in the U251 cell line and measure the expression of miRNAs. We found that miRNA inhibitor significantly reduced the expression of miRNAs compared with transfected NC inhibitor cells (Fig. 10a). We further performed CCK8 to investigate the effect of miRNAs on the viability of glioma cells. Silencing of miR-155-5p, miR-196a-5p, miR-196b-5p, miR-200a-3p, miR-503-5p, and miR-15b-5p delayed the proliferation of U251 cell lines, while silencing of miR-10b-5p or miR-204-5p has no effect (Fig. 10b). Wound healing and transwell assays were employed to explore whether cell migration and invasion were influenced after silencing of miRNAs. We found that the knockdown of miR-155-5p, miR-196a-5p, miR-196b-5p, miR-200a-3p, miR-503-5p, and miR-15b-5p significantly decreased cell migration and invasion compared with the control, while silencing of miR-10b-5p or miR-204-5p has no effect (Fig. 10c–f). Moreover, the knockdown of miR-155-5p, miR-196a-5p miR-196b-5p, miR-200a-3p, miR-503-5p, and miR-15b-5p significantly inhibited the well-known immune evasion markers of *PD-L1*, *CTLA4*, and *FOXP3* at the protein level, while silencing of miR-10b-5p or miR-204-5p has no effect (Fig. 10g and Supplementary Fig. 19). Taken together, these results suggested that miR-155-5p, miR-196a-5p, miR-196b-5p, miR-200a-3p, miR-503-5p, and miR-15b-5p significantly promote cell migration, invasion, proliferation, and are associated with immune evasion in glioma.

## Discussion

MiRNA expression deregulation has been widely observed in cancer development and progression. However, less attention has been paid towards understanding the underlying mechanisms of miRNA dysregulation. In recent years, more and more evidences have indicated the genetic and epigenetic machineries are extensively involved in the process of miRNA biogenesis and impair its expression in cancer cells[57,58]. Here we

focus on two widely-known genetic and epigenetic processes of DNA methylation and copy number variation, in order to get deeper insights into this intricate regulation as well as their roles in cancer development. Different with previous studies which only investigate individual mechanism, we sought to inspect and give a more comprehensive description for both effect of DNAm and CNV on miRNA expression simultaneously. Our analysis not only provides an molecular subtyping system for LGG, but also reveals that the regulatory mechanism surrounding miRNA may be closely related to tumor microenvironment and immune evasion, as well as the prognostic differences of patients. This work indicated that integrative analysis of multi-omics profiling focus on miRNA regulation could identify meaningful molecular subtypes which are in connection with mechanisms and sample heterogeneity in tumors as well as potential biomarkers and therapy targets.

Illustrating the mechanisms involved in transcriptional regulation of miRNAs remains limited, mainly due to that obtaining high-throughput profiles around miRNAs at genetic and epigenetic layers is still challenging. Toward this end, we developed a strategy and successfully extracted miRNA promoter methylation information by using credible miRNA promoter annotation and repurposing the 450K microarray data, which were originally designed to detect the methylation pattern of protein coding genes. Although many miRNA genes reside within intronic regions and are thought to co-transcribed with their host genes, increasing evidences have indicated that intronic miRNAs carry their own promoters, therefore have independent transcription[59]. In such case, we decided to use experimentally verified promoter region for individual miRNAs but without that from host genes being used. In addition, we also obtained miRNA CNV profiling by examining miRNAs *loci* which located in in somatic copy number variation regions. Our results confirmed such associations between DNAm or CNV aberration and miRNA expression. Akin to protein coding genes, the methylation at promoter regions and the expression for corresponding miRNAs are often observed to have negative correlation, whereas for CNVs, the positive correlations dominant for their associations, although few opposite trends also can be observed. We found that about one third of the miRNAs in the human genome tend to be affected by CNV or DNAm alterations, and many of them were driven by a combination of both mechanisms. This phenomenon suggesting the crosstalk between genetic and epigenetic layers are widespread in cancer cell, and may function in maintaining the precise expression of genes.

The concomitant regulation of miRNA expression by genomic CNV and DNAm was also indicted by the genomic distribution of miRNA genes affected. Surprisingly, both the CNV-miRs and DNA-miRs exhibited genomic preference for three different chromosomes, especially the chromosome 14. This concentrated distribution pattern may raise from the expansion of miRNA gene families in the genome. Several studies have indicated that some miRNA families may evolved from genome or tandem duplication events, their family members underwent fast amplification and rapid divergence, thus may have conserved regulatory elements[60]. This could be another clue for elaborating the transcriptional regulation of miRNA family members. This co-regulation pattern in the genome may allow them to regulate the genes within key pathways that involved in cancer, which provides us another perspective to understand the mechanism of miRNA evolution.

Our classification analysis based on CNV and DNAm associated miRNAs revealed molecular features that represent precision therapy

targets and biomarkers for LGG management. Analysis indicated that our subtyping system is superior to those mutational profiling or multi-omics classification methods based on protein-coding genes for prediction of patients' prognosis[61,62]. As an effective gene expression regulator, one miRNA may control dozens to hundreds of target genes. One could hypothesize that disturbance to only a few miRNAs may cause huge downstream runaway effect[63]. In such case, miRNAs show higher value for tumor classification, development of such tools around miRNA multi-omics data may provide us a better alternative for early diagnosis and accurate prognosis predictions in cancers.

Remarkably, our proposed sample classification has a good concordance with the mutation profiles among the subtypes, particularly for *IDH1*, *TP53* and *EGFR*. Somatic mutation is another mechanism that may impact the biogenesis of miRNAs. However, the detailed process remains unrevealed. One inspiring example is the *IDH1*, somatic mutations of this gene are strongly associated with the DNA methylome and transcriptome reorganization in LGG[64]. Whether DNA methylation mediates this indirect association between *IDH1* mutation and miRNA expression remains further analysis. Besides, the association of miRNA expression and mutation frequencies of other genes within LGG remains unknown. Nonetheless, our findings provide potential therapy targets for corresponding subtypes and brings different enlightenment to the study of the intrinsic regulatory mechanisms for LGG.

Analysis of biological characteristics revealed significant differences in the tumor immune microenvironment among molecular subtypes observed. Studies have extensively shown that tumor infiltrating lymphocytes are involved in tumor progression and invasiveness. Infiltrating lymphocytes include various lymphocytes with different activities, particularly the CD8+ and CD4+ T cells, which are usually associate with favorable prognostic in patients[65]. As found in our study, higher level of TILs but the poor survival of IC2 indicated this group as an immunosuppressive subtype. It is worth noting that a recent study based on integrated analysis of DNA methylome and transcriptome data of glioblastoma revealed robust immune subtypes based on lncRNA methylation features. The immune-hot tumor also exhibited immune evasion features and poor survival of the patients, which indicated non-coding RNAs could play essential roles for the immunoregulatory process in glioma[66]. By estimating the correlations with infiltrating immunocytes and immune pathway activities, we identified immune-related miRNAs that may have prognosis value for LGG patients with essential immune miRNAs were prioritized, many of them have been reported to be associated with immune regulation and cancer therapy from previous studies. For instance, miR-15b-5p mediates the degradation of *PD-L1* mRNA through interaction with its 3'-untranslated region, by which to induce CD8+ T and NK cell activation and cytotoxicity against neuroblastoma[67]. Similar observation also applies to miR-155-5p, by targeting *PD-L1*, this miRNA was found to activated CD8+ T cell function and suppress ovarian cancer progression[68]. The prognosis model constructed based on the expression of these miRNAs performs better than other traditional molecular biomarkers and was validated by an independent cohort. These results indicated that the integration of DNAm and CNV is an effective method to study the immune regulation of miRNAs for cancer prognosis.

In summary, we have designed and implemented a strategy to facilitate the integration of copy number variation, DNA methylation and miRNA expression data. Analysis results demonstrate the added value of for molecular subtyping of LGG by applying our method to the multi-omics data. Systematic study also characterized the immune landscape for glioma patients based on miRNAs related multi-omics data and depicted the crosstalk among DNA methylation, copy number variation and miRNA expression for immune regulation and evasion. The repertoire of immune-related miRNAs will facilitate the development of immunotherapeutic targets for glioma in future.

## Methods
### Data collection and preprocessing
The genome-wide profiling data for LGG was download from TCGA (https://portal.gdc.cancer.gov/)[69]. Here we obtained 512 samples with

miRNA expression, 505 samples with DNAm and 515 samples with CNV detection, a total of 500 samples with all these three different types of high-throughput profiling data were retained for further integrative study. The miRNA-seq profiling data which quantified as RPM (reads per million mapped reads) was logarithm transformed (base 2) in order to regularize the data. For DNAm data based on Illumina Infinium HumanMethylation450 BeadChip, methylation level of each probe was measured as the beta-value. We removed the probes whose beta-values are missing in more than 30% of the samples, the remaining probes with missing values were imputed by using the k-nearest neighbors (KNN) method[70]. Finally, the BMIQ method was used to correct for the type II probe bias[71]. For CNV profile, segmentation data generated by Affymetrix SNP 6.0 platform were used. The "nocnv.seg" file contains chromosome location and segment mean value (log2(copy-number/2)) for each sample and was used to capture the focal somatic CNV. In addition, we also collected RNA-seq data for 500 LGG samples which was quantified as log2 transformed FPKM (Fragments Per Kilobase per Million), as well as the somatic mutation data for 508 samples generated by MuTect2 pipeline. Furthermore, clinical information of all samples including age, gender, survival status, histological type, tumor stage and overall survival time was also retrieved from the TCGA data portal.

An external dataset of 198 glioma patients with miRNA expression profiling data was downloaded from the Chinese Glioma Genome Atlas (CGGA)[72]. This miRNA profiling data is based on the Illumina Human v2 MicroRNA Expression BeadChip. We transfer the miRNA ID from the microarray data to that from miRBase v21 by using the "ID convertor" tool in dbDEMC v3.0[73]. The expression values were logarithmically transformed (base 2) and quantile normalized. The associated clinical data which includes survival status and overall survival time of the patients was also obtained.

### Patients in the in-house cohort and sample collection
Human tissue samples of 30 pairs of adjacent non-tumor and glioma samples were collected from the Department of Neuro-surgery of the First Affiliated Hospital, Wannan Medical College (Wuhu, Anhui, P. R. China) and Department of Neuro-surgery of Huashan Hospital, Fudan University (Shanghai, P. R. China) from February 2023 to September 2023. Patients received no systemic treatment before the surgery. All samples obtained at the surgery were directly preserved in liquid nitrogen. Two pathologists evaluated all specimens according to the WHO guidelines. Patient consent was obtained. All surveys and experiments were approved by the Ethic Committee for Clinical Research of the First Affiliated Hospital of Wannan Medical College and the Huashan Hospital of Fudan University, respectively. All ethical regulations relevant to human research participants were followed.

### DNA copy numbers profiling construction for miRNAs
To construct copy number profiling for miRNAs of LGG, we investigated whether the positions of miRNA loci can be mapped to genomic regions with copy number in a certain sample. We first obtained the genomic coordinates for mature miRNAs via miRBase v21[74]. We assigned the locus of each miRNA to the copy number values by using the GISTIC2[75]. Then segment-mean values in distinct genomic loci (genome assembly hg38) were extracted from TCGA SNP array data. In this way, the copy number profiling data for individual estimates were obtained for a total of 2265 miRNAs.

### miRNA promoter identification and Infinium 450K array probe reannotation to construct miRNA methylation profiles
To characterize DNA methylation patterns for miRNAs, we employed a strategy to reannotate the probes of Infinium 450K arrays into miRNA-associated promoter regions. We first obtained genome coordinates for miRNA promoters from the fifth edition of the Functional Annotation of Mammalian Genome database (FANTOM5)[76]. FANTOM5 incorporates miRNA transcription start site (TSS) information generated by Cap Analysis Gene Expression (CAGE) methods, which was based on genome

assembly hg19. The coordinates of miRNA TSS were converted to genome assembly hg38 by using the liftOver tool from the UCSC genome browser. As a result, a total of 2050 mature miRNAs are identified for unique physical genomic location of TSS among 2587 miRNAs from miRBase v21. Here we define the promoter region for individual miRNA as between 2000 bp upstream to 500 bp downstream of TSS, which is akin to protein coding genes. By matching the genomic coordinates of the miRNA TSS with the 450K Array probe locations, we filtered 8825 cpg sites that are located within miRNA promoters. Then we assign the average methylation value of the probes within promoter region for each miRNA to obtain its final methylation value. Finally, we obtained methylation profile for 1977 miRNAs for LGG.

### Identification of CNV and DNAm related miRNA sets
The Pearson correlation coefficients ($r$) were calculated for paired data of CNV and miRNA-seq and between DNAm and miRNA-seq for each miRNA, respectively, the correlation coefficient was then transformed to the Fisher $Z$-scale according to the formula:

$$Fisher\ Z = \frac{1}{2} ln \frac{1+r}{1-r}$$

The miRNA of FDR adjusted $p$ value $< 0.05$ with correlation test constitutes the miRNA set significantly positively related to CNV (CNV-miRs) and the miRNA set negatively related to methylation (DNAm-miRs), respectively. We further used a linear regression model to dissected the influence of DNAm and CNV on miRNA expression and to investigate the core functions of the two groups of miRNAs. Specifically, we propose expression level of specific miRNA as $Y$ ($Y_{miR}$), and its methylation level as $M$ ($M_{miR}$), as well as the copy number level as $N$ ($N_{miR}$), we infer its association coefficients with the DNAm ($\beta_m$) and CNV ($\beta_n$) for each miRNA across all samples as:

$$Y_{miR} \approx \beta_0 + \beta_m * M_{miR} + \beta_n * N_{miR}$$

We filtered those miRNAs with negative coefficients for DNAm and positive coefficients for CNV under FDR $< 0.05$. In this way, we identified the core subsets of DNAm-miRs and CNV-miRs that uniquely influenced by these two different mechanisms, respectively.

### Clustering analysis of multi-layered genomic profiles
Nonnegative matrix factorization (NMF) is an unsupervised clustering method which was widely used for tumor subtype identification. Here we employ NMF to identify stable sample clusters by using the expression of CNV-miRs and DNAm-miRs, respectively. Specifically, cluster analysis with standard "brunet" method and 50 iterations was used, we set the number of cluster $k$ as 2–7, the optimal cluster number $k$ was then determined based on the observed consensus map together with cophenetic correlation between clusters. The mean silhouette width was computed by R package "NMF" to examine consensus membership matrix[77]. In addition, we also applied "MOVICS" R package to integrate the CNV data, methylation data, as well as the expression profile data from both CNV-miRs and DNAm-miRs for the integrative analysis of cancer subtyping[78]. MOVICS provides an interface for 10 state-of-the-art multi-omics data clustering algorithms (SNF, PINSPlus, NEMO, COCA, LRAcluster, CIMLR, Consensus clustering, IntNMF, MoCluster, iClusterBayes) and combines the output of each algorithm to obtain more robust classification. We calculated the clustering prediction index to find the optimal number of clusters and the consensus matrix was used to validate robust clustering of the samples.

### Measurement for immune cell infiltration and antitumor immunoactivity
We identified and evaluated the immune infiltrates levels by the Tumor immune estimation resource (TIMER) algorithm, which is used for systematic estimation of the abundances of six immune cell types (B cell, CD4+ T cell, CD8+ T cell, neutrophil, macrophage and dendritic cell) based on transcriptome profiles[79]. The abundances of these cells were compared across different molecular subtypes for quantitative characterization of the tumor immune microenvironment. We also used ESTIMATE (Estimation of STromal and Immune cells in MAlignant Tumor tissues using Expression data) to assess immune activity of the tumor samples. ESTIMATE yields the immune score and stromal score for each sample that quantifies the levels of infiltration levels of immune and stromal cell based on its gene expression profiles[36]. We collected 17 immune relevant pathways and the associated 1811 genes from ImmPort[35]. In addition, three different indices were calculated to measure the immune activities for tumor samples: Cytolytic activity (CYT) score, Major Histocompatibility Complex (MHC) score and Cytotoxic T Lymphocyte (CTL) score. The CYT score was defined as the geometric mean of $GZMA$ and $PRF1$:

$$Score_{CYT} = \sqrt{Exp_{GZMA} * Exp_{PRF1}}$$

These two genes are upregulated in activated CD8+ T cells and play important roles in cytolysis and response to $PD-1$ and $CTLA4$ related immunotherapy[80,81]. The MHC score was calculated as the average expression as nine genes of $HLA-A$, $PSMB9$, $HLA-B$, $PSMB8$, $HLA-C$, $B2M$, $TAP2$, $NLRC5$ and $TAP1$:

$$Score_{MHC} = \sum Exp_n / 9$$

These genes were identified as the core set of MHC-I and are associated with activity of MHC-I antigen presentation[82]. The CTL score for each sample was calculated as average expression from five genes of $CD8A$, $CD8B$, $GZMA$, $GAMB$ and $PRF1$:

$$Score_{CTL} = (Exp_{CD8A} + Exp_{CD8B} + Exp_{GZMA} + Exp_{GZMB} + Exp_{PRF1})/5$$

These five genes are important factors to measure cytotoxicity of tumor-infiltrating T cells and immune cell effector function[83]. In addition, we also obtained the gene sets that represents different immune signatures from several publications, including the immune checkpoint, human leukocyte antigen (HLA), interferon (IFN) response and tumor-infiltrating lymphocytes (TILs), which are collected from publication of Ju et al.[84], and the gene sets for Macrophage/Monocyte, TGF-β response, IFN-γ response, and Wound healing that collected from Vesteinn et al.[85]. We use the single-sample Gene Set Enrichment Analysis (ssGSEA) algorithm to estimate the activity of tumor samples by the immune signatures obtained[86]. Finally, another 63 immune cell type specific gene markers for 21 immune cell populations were collected from TISIDB[87].

### Immune evasion and immunotherapy response evaluation
To evaluate complex interactions among cancer cells and immune microenvironment, we constructed the scoring system following Shuichi et al. to describe factors that relevant to the tumor proliferation (T-factor), antitumor immunity (I-factor) and immunosuppression (S-factor)[41]. Briefly, nine representative gene sets related to those factors were collected from the molecular signature database (MsigDB V.7.1)[40], which include Gene Ontology (GO) terms of "SPINDLE_LOCALIZATION" and "POSITIVE_REGULATION_OF_GLUCOSE_METABOLIC_PROCESS" for T-factor, "T cell", "B cell" Signatures and GO terms of "MHC_PROTEIN_COMPLEX" for "I-factor", and "NEGATIVE_REGULATION_OF_CYTOKINE_PRODUCTION_INVOLVED_IN_INFLAMMATORY_RESPONSE", "POSITIVE_REGULATION_OF_CELL_MIGRATION_INVOLVED_IN_SPROUTING_ANGIOGENESIS", "NEGATIVE_REGULATION_OF_HYPOXIA_INDUCED_INTRINSIC_APOPTOTIC_SIGNALING_PATHWAY" as well as "EPITHELIAL_TO_MESENCHYMAL_TRANSITION" for "S-factor", the ssGSEA enrichment score was calculated and z-score normalized, the mean values of z-scores for the gene sets allocated to T-factors, I-factors or S-factors and scored was calculated to obtain T-scores, I-scores or S-scores.

Finally, we apply the Tumor Immune Dysfunction and Exclusion (TIDE) algorithm to estimate the immunotherapy response of LGG patients[50]. Higher TIDE scores implicated a higher potential of tumor immune evasion, thus less likely to benefit from immunotherapy.

### Immune-related miRNAs identification and prognosis model development

To identify miRNAs associated with immune regulation, Pearson correlation analysis was performed between miRNAs expression and immune cell infiltrating level across samples. miRNAs with FDR less than 0.05 were considered as significant. We also collected the public available immune-related miRNAs from the RNA2Immune database for functional validation[38]. This database provides a high-quality experimentally supported resource that links ncRNA regulatory mechanisms to four sections of "immune cell function", "immune diseases", "cancer immunity", and vaccines. We selected miRNAs from the "immune cell function", and "cancer immunity" for validation propose. To find the miRNAs with the height prognostic value, we applied Cox-proportional hazards analysis by using the Least absolute shrinkage and selection operator (LASSO) estimation. Among the immune miRNAs that were significant correlated with immune cell infiltrating, key immune miRNAs were selected by performing the LASSO-penalized Cox regression, which was implemented by the glmnet R package (version 4.1-7). Finally, an immuno-miRNA related prognostic model was constructed by utilizing the regression coefficients derived from Cox regression analysis to multiply the normalized expression level of each immune miRNA as follows:

$$
\begin{aligned}
\text{Risk Score} = & (0.1617 * Exp_{miR-10b-5p}) + (0.3419 * Exp_{miR-155-5p}) \\
& + (0.0695 * Exp_{miR-196a-5p}) + (0.1391 * Exp_{miR-196b-5p}) \\
& + (0.1474 * Exp_{miR-200a-3p}) + (0.0840 * Exp_{miR-204v5p}) \\
& + (0.0996 * Exp_{miR-503-5p}) + (0.2083 * Exp_{miR-15b-5p})
\end{aligned}
$$

The "surv_cutpoint" function within the "survminer" package was applied to determine the best cutoff to classify patients into high-risk and low-risk groups. The Kaplan–Meier survival analysis and log-rank test were used to assess the predictive ability of the prognostic model.

### Validation of the prognostic model from independent cohorts

To validate whether the predictions of the prognostic model were independent of other clinical features, we collected miRNA profiling data form CGGA and calculated the risk scores for the patients by using the model constructed. In addition, the predictive accuracies were compared using the time-dependent receiver operating characteristic (ROC) analyses and the associated area under curve (AUC) score.

### Construction and evaluation of the nomogram

We assessed the correlation between prognostic model and other clinicopathological features. The Cox proportional hazards model was used to perform standard univariate and multivariate analysis. Prediction error curves were used to compare the accuracy of survival models. The Cox regression coefficients were used to construct the nomogram. Calibration plots were generated to explore the performance of the nomogram. Time dependent ROC analysis was performed to assess the predictive accuracy of the nomogram. Decision curve analysis was used to assess the clinical practicability of the nomogram. The statistical significance was set at 0.05.

### miRNA target prediction and functional enrichment analysis

For both miRNAs and protein coding genes that differentially expressed between groups, the empirical Bayes framework of moderated *t*-statistics were employed for assessing differential expression in microarray experiments[88]. miRNA target prediction results were obtained from databases of TargetScan[89] and miRDB[90]. The miRNA-target interactions predicted by both of the two algorithms were identified as valid ones. In addition, we also collected experimentally validated miRNA targets from

miRTarBase[91]. For functional annotation of the miRNA targets involved in the regulatory network, The GO enrichment analysis was performed by using clusterProfiler[92]. In addition, hallmark gene sets were obtained from MsigDB, and Gene Set Variation Analysis (GSVA) analysis was used to infer the gene set/pathway activity in individual sample

### Distinct genomic features across different groups

The tumor mutation burden (TMB) and oncoplot of somatic mutation data were calculated and generated by using R package "maftools"[93]. The different mutational frequencies for individual genes between groups were calculated by chi-square test, the *p* value that less than 0.05 were identified as having achieved statistical significance. In addition, the druggable pathways and categories are also obtained by enrichment analysis.

### scRNA-seq data processing and analysis

The scRNA-seq data of 10 *IDH1* mutant and 8 *IDH1* wild-type glioma samples were downloaded from the GEO database (GSE152273). The R package "Seurat" was used to convert scRNA-seq data into Seurat objects[94]. We first performed quality control for the scRNA-seq data by removing cells with the number of genes mapped less than 50 and cells with more than 5% of mitochondrial genes. Then the data was normalized by "NormalizeData" function. The Harmony R package was employed for batch correction[95]. The top principal components (PC) were extracted by principal component analysis (PCA) based on the top 2000 highly variable genes. Then, the results were visualized by uniform manifold approximation and projection (UMAP) methods on a two-dimensional map. Subsequently, the "SingleR" package was used to annotate the cell subpopulations of the different clusters[96]. Finally, we aggregate cells from each sample to create pseudobulk expression profiles using the "AggregateExpression" function.

### Quantitative real-time polymerase chain reaction (qRT–PCR)

Total RNA was extracted from tissues with TRIzol reagent from 30 pairs of LGG and normal tissues according to the manufacturer's protocol. Reverse transcription was performed using the PrimeScript RT Reagent Kit. For miRNA quantification analysis, a Bulge-Loop™ miRNA qRT-PCR primers set was designed and synthesized (RiboBio, Guangzhou, China). The quantitative PCR assay was performed using the SYBR Green PCR kit. The expression of eight miRNAs of the in-house cohort was normalized by GAPDH, acting as an internal control. Relative miRNA expression levels were calculated by the $2^{-\Delta\Delta CT}$ method. Each sample was repeated in triplicate and analyzed using relative quantification method.

### Cell culture and transfection

The human glioma cell line U251 used in this study was purchased from the American Type Culture Collection (ATCC, Manassas, VA, USA). DNA fingerprinting, cell vitality detection, isozyme detection, and mycoplasma detection were used to characterize all cell lines. The cell line was regularly cultured in Dulbecco's modified Eagle media (DMEM; Gibco; Thermo Fisher Scientific; Waltham, MA, USA) supplemented with 10% fetal bovine serum (FBS; Gibco; Thermo Fisher Scientific) at 37 °C in a humid environment with 5% $CO_2$.

Transfection of miRNA inhibitors were performed using Hieff TransTM in vitro siRNA/miRNA Transfection Reagent (Yeasen, China) according to the manufacturer's instructions. U251 cells were plated in T-25 cells culture flasks and 96 wells plates, 24 wells plates, and transfected with miRNA inhibitors. At 48–72 h after transfection, the cells were collected and used for experiments.

### Western blotting

The membrane was incubated with the following primary antibodies at 4 °C overnight after being blocked with 5% non-fat milk at room temperature for 1 h: anti-PD-L1 (1:1000, Cat#13684; Cell Signaling Technology, Danvers, MA, USA), anti-β-actin (1:1000, Cat#A1978, Sigma-Aldrich, Victoria, BC, Canada), anti-CTLA4 (Cat#53560; Cell Signaling Technology, Danvers, MA, USA), and anti-FOXP3 (1:1000, Cat#12632; Cell Signaling

Technology, Danvers, MA, USA) antibodies. The membrane was washed three times, followed by treatment with horseradish peroxidase-conjugated anti-mouse or anti-rabbit secondary antibody. After that, a Tanon 5200 system (Tanon, Shanghai, PR China) was used to photograph and wash the membrane. Using Quantity One software from Tanon (Shanghai, PR China), the intensity of protein bands was assessed using densitometric analysis. Target gene protein expression was calibrated to β-actin protein expression. A triplicate of the experiment was run.

### CCK-8 assay
Cells were seeded, cultivated for an overnight period, and then given the CCK8 solution in 96-well plates. The cells were then treated with the CCK-8 reagent at 37 °C in a humid environment that contained 5% $CO_2$. The absorbance at 450 nm was measured on an enzyme analyzer (Tecan infinite M2009PR, Tecan, Männedorf, Switzerland). The experiment was carried out three times.

### Transwell assay
Cell migration and invasion were performed with Transwell chambers (BD Biosciences, USA). After 48 h of corresponding treatment, U251 cells in each group were inoculated into the upper chamber of the Transwell chamber at the density of $4 \times 10^4$ cells/well (100 μl culture medium containing 5% fetal bovine serum). Besides, 500 μl culture medium containing 10% fetal bovine serum was added to the 24-well culture plate of the lower chamber. After 24 h of routine culture, the chamber was removed, and cells from the upper layer of microporous membranes were wiped with cotton swabs. After that, cells were immobilized in 4% paraformaldehyde solution for 10 min at room temperature and stained with 0.5% crystal violet solution (Sigma-Aldrich; Merck KGaA) for 15 min. In the last step, 5 visual fields were randomly selected and observed under an optical microscope (Nikon, Japan) to count the number of cells invading the sub-layer of the microporous membranes of the chamber.

### Wound healing assay
Cell migration was observed using a wound-healing assay. When the transfected U251 cells were maintained in a 6-well plate, achieving 90–95% confluence, scratches were generated using the micropipette tips. The wound state was observed at 0 and 24 h after scratching with an X71 inverted microscope (Olympus, Tokyo, Japan).

### Statistics and reproducibility
All the analyses were performed on TCGA LGG dataset ($n = 500$) unless mentioned otherwise. For all differential analyses, altered features were investigated with respect to corresponding groups (IC1: $n = 211$, IC2: $n = 75$, IC3: $n = 142$, IC4: $n = 72$). The significance of differences between two groups was determined by Wilcoxon rank sum test. The frequency differences among different groups were determined by using chi-square test. For patient survival analyses, Kaplan–Meier plots were created, we used the Cox proportional hazard model and the log-rank test to determine the difference of survival between different groups. Unless stated otherwise, all statistical tests were performed by R software (version: 4.1.3). Statistical significance was assessed by calculating $p$ values using various statistical methods. A $p$ value < 0.05 was considered statistically significant. If necessary, $p$ values were corrected for multiple tests with the Benjamini–Hochberg procedure to calculate false discovery rate (FDR).

### Reporting summary
Further information on research design is available in the Nature Portfolio Reporting Summary linked to this article.

### Data availability
Data analyzed in this manuscript are already publicly available from The Cancer Genome Atlas (TCGA) data portal: https://portal.gdc.cancer.gov/ and the Chinese Glioma Genome Atlas (CGGA) data portal: http://www.cgga.org.cn/. The Count matrix of single cell RNA-seq data was obtained from the GEO database with the accession number GSE152273. The source data behind the graphs in the paper are provided in Figshare: https://doi.org/10.6084/m9.figshare.25989109.v1[97].

### Code availability
In this study, we used software general workflow codes, without generating any new code. R packages and specific functions, as well as softwares used are described in relevant sections in the "Method" section.

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

## Acknowledgements

The study was supported by the National Natural Science Foundation of China (91959106, 32071270, 82072370, 82103376); Outstanding Innovative Research Team for Molecular Enzymology and Detection in Anhui Provincial Universities (2022AH010012); Program for Excellent Sci-tech Innovation Teams of Universities in Anhui Province (2022AH010074); the Major Science and Technology Projects in Anhui Province (202003a06020009); Anhui Natural Science Foundation (2208085MC48); Key University Science Research Project of Anhui Province (2023AH051746, 2023AH051768); Climbing Peak Training Program for Innovative Technology team of Yijishan Hospital, Wannan Medical College (PF201904); Peak Training Program for Scientific Research of Yijishan Hospital, Wannan Medical College (GF2019T01, GF2019G15); the Open Project of Key Laboratory of Non-coding RNA Transformation Research of Anhui Higher Education Institution, Wannan Medical College (RNA202205); Science and Technology Application Basic Research Project of Wuhu (2022jc60); and Key Research and Development Foundation supported by Science and Technology Department of Sichuan Province (2023YFS0243). The sponsors have no role in study design, in the collection, analysis and interpretation of data, in the writing of the report, and in the decision to submit the article for publication.

## Author contributions

Z.Y., K.L. and H.Y. conceived the study. X.L. and H.X. performed experimental validation analysis. A.E.T. supervised data analysis. L.X. and JY.L. performed data analysis. M.F., JU.L., H.Z., Y.W., L.Z. and Y.H. collected and processed samples. Z.Y. wrote the paper. A.E.T. revised the paper. All authors read and approved the paper.

## Competing interests

The authors declare no competing interests.

## Additional information

[1]Center for Medical Research and Innovation of Pudong Hospital, and Intelligent Medicine Institute, Shanghai Medical College, Fudan University, 131 Dongan Road, Shanghai 200032, China. [2]Department of Nuclear Medicine, The First Affiliated Hospital of Wannan Medical College (Yijishan Hospital of Wannan Medical College), Wuhu 241001 Anhui, China. [3]Anhui Province Key Laboratory of Non-Coding RNA Basic and Clinical Transformation, Wuhu 241001 Anhui, China. [4]Key Laboratory of Non-Coding RNA Transformation Research of Anhui Higher Education Institution, Wannan Medical College, Wuhu 241001 Anhui, China. [5]Department of Neurosurgery, Huashan Hospital, Fudan University, Shanghai 200040, China. [6]Neurosurgical Institute of Fudan University, Shanghai Key Laboratory of Brain Function Restoration and Neural Regeneration, Shanghai 200040, China. [7]CAS Key Laboratory of Computational Biology, Shanghai Institute of Nutrition and Health, University of Chinese Academy of Sciences, Chinese Academy of Sciences, 320 Yue Yang Road, Shanghai 200031, China. [8]Emergency Department, West China Hospital, West China School of Medicine, Sichuan University, Chengdu, Sichuan, China. [9]Department of Medical Cosmetology, Beijing Tiantan Hospital, Capital Medical University, 100070 Beijing, China. [10]Department of Gastrointestinal Surgery, The First Affiliated Hospital of Wannan Medical College (Yijishan Hospital of Wannan Medical College), Wuhu 241001 Anhui, China. [11]Central Laboratory, The First Affiliated Hospital of Wannan Medical College (Yijishan Hospital of Wannan Medical College), Wuhu 241001 Anhui, China. [12]Shanghai Fifth People's Hospital, and Intelligent Medicine Institute, Shanghai Medical College, Fudan University, 131 Dongan Road, Shanghai 200032, China. [13]These authors contributed equally: Zhen Yang, Xiaocen Liu, Hao Xu.
✉e-mail: zhenyang@fudan.edu.cn; lvkun315@126.com; yanghui0203@wnmc.edu.cn

