## [Peer Review File · Communications Biology]

Reviewers' comments:

Reviewer #1 (Remarks to the Author):

Yang et al. integrated the copy number variation and DNA methylation to construct miRNA regulatory networks, which could reveal tumor heterogeneity and facilitate immune evasion in LGG. The study is well designed, however, several problems should be addressed before publication.

1. In the introduction section, the author said 'Transcriptional disorders caused by CNV changes have been identified as driving events in LGG progression', but related references are missing. In the following introduction about DNA methylation and LGG cancer, the author cited several unrelated studies including mouse development and endometrial cancer (ref. 22-25). Authors should introduce the linking between DNA methylation and carcinogenesis of glioma like PMID: 38453529, 37494934...
2. The difference of the proportional frequencies in each category of DNA methylation probes for DNAm miRNAs and all miRNAs should be explored via the chi-square test in Figure 1D-E.
3. The description of the formula including the association coefficients between miRNA expression and CNV/DNAm, and the risk score should move from the results to the methods section.
4. Since there is a significant overlap between the CNV and DNAm-related miRNA set, it's necessary to remove these overlapped miRNAs to identify LGG molecular subtypes in Figure.
5. Why do authors only perform DE gene analysis for IC2 and other clusters? Since authors want to characterize the immune landscape of LGG subtypes, it's necessary to perform DE genes analysis for each cluster compared to other clusters and determine whether immune pathways changed in the specific cluster.
6. The immune-related analysis this study performed is similar to the Xu et al. study (Theranostics,2013, PMID:37056564), which reveals the glioma immune heterogeneity via the DNA methylation of lncRNAs. The study found that immune activity, immune signature, and immune cell infiltrations exhibited significant differences between glioma subtypes. Similarly, the immune-hot phenotype also exhibited immune evasion features and poor survival. The authors should discuss the results compared to this study in the discussion section.
7. Apart from PD-L1, there are several essential genes (like KLRB1, CD70, FASLG...) that mediated the immune evasion of glioma patients. Authors should also identify the miRNA interactions with these genes.

Reviewer #2 (Remarks to the Author):

The authors present an interesting study on miRNA regulatory network in lower grade glioma (LGG). LGG is the most common and aggressive type of brain tumor, representing 40% of all brain tumors. At present, there is no robust biomarker or subtyping for the prognosis of LGG, thus, the current study on miRNA-based biomarker discovery is of high clinical significance. To this end, the authors performed an integrative study by systematically analyzing multi-omics datasets from TCGA for LGG, including DNA methylation, copy number variation, and miRNA expression, to uncover miRNA-related dysregulation in LGG. In doing so, they revealed a large number of miRNAs that were altered in tumors. Further miRNA target analysis also sheds light on miRNA-mediated networks that are dysregulated in tumor progression and response to immunotherapy. Importantly, major findings from the bioinformatics analysis were also experimentally validated using an independent cohort as well as glioma cell lines. Thus, results from this study not only identified potential biomarkers for LGG prognosis, but also suggested novel gene targets and pathways that are relevant to novel therapeutic development for LGG. Overall, this is a well designed and executed study focusing on an important clinical problem. However, there are a few minor issues that should be addressed.

- 1) In Fig. 1D, the distribution of DNA methylation probes is presented. However, this result was not normalized by the length of different genic regions, and thus it is hard to interpret its biological

significance.

2) The authors suggested an interesting interplay between miRNA dysregulation and changes in the tumor immune microenvironment. The cluster-specific immune profiles are particularly intriguing. I am wondering whether these findings can be experimentally validated by immunostaining of the 30 in-house LGGs.

Reviewers' comments:

Reviewer #1 (Remarks to the Author):

Yang et al. integrated the copy number variation and DNA methylation to construct miRNA regulatory networks, which could reveal tumor heterogeneity and facilitate immune evasion in LGG. The study is well designed, however, several problems should be addressed before publication.

Response: We are grateful for the reviewer's positive feedback and insightful suggestions.

1. In the introduction section, the author said 'Transcriptional disorders caused by CNV changes have been identified as driving events in LGG progression', but related references are missing. In the following introduction about DNA methylation and LGG cancer, the author cited several unrelated studies including mouse development and endometrial cancer (ref. 22-25). Authors should introduce the linking between DNA methylation and carcinogenesis of glioma like PMID: 38453529, 37494934...

Response: We thank for the suggestions, as reviewer suggested, we have added the references (Ref 22) for the description of the associations between CNV and LGG. In addition, we have also added the introduction to the linking between DNA methylation and carcinogenesis of glioma and the associated references (Ref 23 and Ref 24)

2. The difference of the proportional frequencies in each category of DNA methylation probes for DNAm miRNAs and all miRNAs should be explored via the chi-square test in Figure 1D-E.

Response: As reviewer suggested, the differences of proportional frequencies in each category of DNA methylation probes for DNAm miRNAs and all miRNAs were evaluated by the chi-square test, the results indicated that there are significant differences for the frequencies between both groups of DNA methylation probes.

3. The description of the formula including the association coefficients between miRNA expression and CNV/DNAm, and the risk score should move from the results to the methods section.

Response: We thank for the suggestion and have moved the description of the two

formulas to the Method Section.

4. Since there is a significant overlap between the CNV and DNAm-related miRNA set, it's necessary to remove these overlapped miRNAs to identify LGG molecular subtypes in Figure.

Response: We thank the reviewer for raising this point. As indicated in our manuscript, we first investigated the functions of the two different groups of miRNAs, as there is significant overlap between two groups of miRNA sets, we used the multiple linear regression to dissect the DNAm- and CNV-miRNA expression associations. In this way, we identified the “core subset” of DNAm-miRs and CNV-miRs that uniquely influenced by the two different mechanisms. We believe it is necessary to remove those overlapped miRNAs if we want investigate their functions precisely as we need to rule out the influences of other mechanisms. Next, we performed prognosis analysis based on the expression of DNAm-miRs and CNV-miRs, respectively, the results showed that both groups of miRNAs presented certain value in predicting patient prognosis, but neither of them show much better effect compared with traditional molecular markers (Figure 2). This indicated that more informative miRNA should be included in the analysis to get better results. The subsequent integrative analysis based on all DNAm-miRs and CNV-miRs confirms this assumption, and much better results for patient's prognosis prediction was obtained (Figure 3B). In this case, we decided to use all the miRNAs as they are informative for prognosis prediction.

5. Why do authors only perform DE gene analysis for IC2 and other clusters? Since authors want to characterize the immune landscape of LGG subtypes, it's necessary to perform DE genes analysis for each cluster compared to other clusters and determine whether immune pathways changed in the specific cluster.

Response: As the reviewer request, we performed differential expression analysis for each cluster compared to other clusters and determined specific immune pathways changed in each specific cluster (Fig S11). Here we focused on exploring the underlying mechanisms for the poor prognosis of IC2 patients, so we keep the IC2 result in the main figure and the results for other clusters in supplementary file.

6. The immune-related analysis this study performed is similar to the Xu et al. study

(Theranostics,2013, PMID:37056564), which reveals the glioma immune heterogeneity via the DNA methylation of lncRNAs. The study found that immune activity, immune signature, and immune cell infiltrations exhibited significant differences between glioma subtypes. Similarly, the immune-hot phenotype also exhibited immune evasion features and poor survival. The authors should discuss the results compared to this study in the discussion section.

Response: We thank for the kindly remind of the reviewer, we have added some discussion about the ncRNA regulation and immune evasion in LGG, and this paper was added in the reference.

7. Apart from PD-L1, there are several essential genes (like KLRB1, CD70, FASLG...) that mediated the immune evasion of glioma patients. Authors should also identify the miRNA interactions with these genes.

Response: We thank for the suggestion. As reviewer suggested, we checked expression of these immune evasion related genes and found that all of them present highest expression in IC2 cluster, which indicated they play critical roles for the immune evasion regulation in LGG. In addition, the miRNAs that have interactions with these genes were also identified. The results were presented as Figure S17.

Reviewer #2 (Remarks to the Author):

The authors present an interesting study on miRNA regulatory network in lower grade glioma (LGG). LGG is the most common and aggressive type of brain tumor, representing 40% of all brain tumors. At present, there is no robust biomarker or subtyping for the prognosis of LGG, thus, the current study on miRNA-based biomarker discovery is of high clinical significance. To this end, the authors performed an integrative study by systematically analyzing multi-omics datasets from TCGA for LGG, including DNA methylation, copy number variation, and miRNA expression, to uncover miRNA-related dysregulation in LGG. In doing so, they revealed a large number of miRNAs that were altered in tumors. Further miRNA target analysis also sheds light on miRNA-mediated networks that are dysregulated in tumor progression and response to immunotherapy. Importantly, major findings from the bioinformatics

analysis were also experimentally validated using an independent cohort as well as glioma cell lines. Thus, results from this study not only identified potential biomarkers for LGG prognosis, but also suggested novel gene targets and pathways that are relevant to novel therapeutic development for LGG. Overall, this is a well designed and executed study focusing on an important clinical problem. However, there are a few minor issues that should be addressed.

Response: We are thankful for the reviewer's positive feedback and appreciate these suggestions.

1) In Fig. 1D, the distribution of DNA methylation probes is presented. However, this result was not normalized by the length of different genic regions, and thus it is hard to interpret its biological significance.

Response: We should point out that as CpG sites are randomly distributed in the genome, it is not very important to consider the length of different genomic regions. The purpose of the analysis here is to compare whether there are distribution differences of the CpG sites associated with miRNA and all CpG sites in the genome, so as to determine whether relevant miRNAs are more likely to be affected by DNA methylation regulation. Therefore, we need to compare the distribution status of different sites by using Chi-square test, as pointed out by the other reviewer, and we have We have made supplementation for this analysis in our paper.

2) The authors suggested an interesting interplay between miRNA dysregulation and changes in the tumor immune microenvironment. The cluster-specific immune profiles are particularly intriguing. I am wondering whether these findings can be experimentally validated by immunostaining of the 30 in-house LGGs.

Response: We thank for this insightful suggestion. However, due to the limited miRNA expression information in the relevant samples and subtyping analysis is unavailable, therefore, we adopted another strategy to conduct this validation analysis. We obtained the single cell RNA-seq data from glioma samples with or without IDH1 mutation, as the IC2 group is more concentrated in IDH wild-type patients, while the other groups are more concentrated in IDH mutant patients, we compared the immune cell fractions and the expression of four immune evasion related genes including *PD-L1*, *KLRB1*, *CD70*, and *FASLG*. The results indicated that IDH1 wild type samples present higher

proportion of immune cells, in addition, three immune evasion related genes including *KLRB1*, *CD70*, and *FASLG* present higher expression in IDH wild type samples. In this way, we provide further validation for our observation about the immune status in different LGG subtypes.

REVIEWERS' COMMENTS:

Reviewer #1 (Remarks to the Author):

All issues have been addressed. I recommended to accept.

Reviewer #2 (Remarks to the Author):

The authors have addressed my previous concerns.